# Spectroscopic evidence for large aspherical β-NAT particles involved in denitrification in the December 2011 Arctic stratosphere

Wolfgang Woiwode[1], Michael Höpfner[1], Lei Bi[2,*], Michael C. Pitts[3], Lamont R. Poole[4], Hermann Oelhaf[1], Sergej Molleker[5], Stephan Borrmann[5,6], Marcus Klingebiel[6,**], Gennady Belyaev[7], Andreas Ebersoldt[8], Sabine Griessbach[9], Jens-Uwe Grooß[10], Thomas Gulde[1], Martina Krämer[10], Guido Maucher[1], Christof Piesch[1], Christian Rolf[10], Christian Sartorius[1], Reinhold Spang[10], Johannes Orphal[1]

[1]Institute of Meteorology and Climate Research, Karlsruhe Institute of Technology, Karlsruhe, Germany
[2]Department of Atmospheric Sciences, Texas A&M University, College Station, TX 77843, USA
[3]NASA Langley Research Center, Hampton, Virginia, 23681, USA
[4]Science Systems and Applications, Incorporated, Hampton, Virginia, 23666, USA
[5]Max Planck Institute for Chemistry, Particle Chemistry Department, Mainz, Germany
[6]Institute for Physics of the Atmosphere (IPA), University of Mainz, Mainz, Germany
[7]Myasishchev Design Bureau, Zhukovsky-5, Moscow Region, Russia
[8]Institute for Data Processing and Electronics, Karlsruhe Institute of Technology, Karlsruhe, Germany
[9]Jülich Supercomputing Centre (JSC), Forschungszentrum Jülich GmbH, 52425 Jülich, Germany
[10]Institute of Energy and Climate Research (IEK-7), Forschungszentrum Jülich GmbH, 52425 Jülich, Germany
*now at: School of Earth Sciences, Zhejiang University, Hangzhou, China, 310027
**now at: Max Planck Institute for Meteorology, Hamburg, Germany

*Correspondence to*: W. Woiwode (wolfgang.woiwode@kit.edu)

**Abstract.** We analyse polar stratospheric cloud (PSC) signatures in airborne MIPAS-STR (Michelson Interferometer for Passive Atmospheric Sounding – STRatospheric aircraft) observations in the spectral regions from 725 to 990 $cm^{-1}$ and 1150 to 1350 $cm^{-1}$ under conditions suitable for the existence of nitric acid trihydrate (NAT) above northern Scandinavia on 11 December 2011. The high resolution infrared limb emission spectra of MIPAS-STR show a characteristic "shoulder-like" signature in the spectral region around 820 $cm^{-1}$, which is attributed to the $\nu_2$ symmetric deformation mode of $NO_3^-$ in β-NAT. Using radiative transfer calculations involving Mie and T-Matrix methods, the spectral signatures of spherical and aspherical particles are simulated. The simulations are constrained using collocated in-situ particle measurements. Simulations assuming highly aspherical spheroids with aspect ratios (AR) of 0.1 or 10.0 and a lognormal particle mode with a mode radius of 4.8 µm reproduce the observed spectra to a high degree. A smaller lognormal mode with a mode radius of 2.0 µm, which was also taken into account, plays only a minor role. Within the scenarios analyzed, the best overall agreement is found for elongated spheroids with AR=0.1. Simulations of spherical particles and spheroids with AR=0.5 and 2.0 return results very similar to each other and do not allow to reproduce the signature around 820 $cm^{-1}$. The observed "shoulder-like" signature is explained by the combination of the absorption/emission and scattering characteristics of large highly aspherical β-NAT particles. The size distribution supported by our results corresponds with ~9 ppbv of gas-phase equivalent $HNO_3$ at the flight altitude of ~18.5 km. The results are compared with the size distributions derived from the in-situ observations, a corresponding Chemical Lagrangian Model of the Stratosphere (CLaMS) simulation, and excess gas-phase $HNO_3$ observed in a nitrification layer

directly below the observed PSC. The presented results suggest that large highly aspherical β-NAT particles involved in denitrification of the polar stratosphere can be identified by means of passive infrared limb emission measurements.

## 1 Introduction

Nitric acid trihydrate (NAT) particles are known to be involved in denitrification and vertical downward transport of $HNO_3$ in the Arctic and Antarctic winter stratosphere and thereby affect polar ozone chemistry (e.g. Waibel et al., 1999; Carslaw et al., 2002.; Grooß et al., 2014). β-NAT is the only nitric acid hydrate known to be thermodynamically stable in condensed state under the conditions of the polar winter stratosphere at temperatures below ~195 K (Hanson and Mauersberger, 1988). The metastable modification α-NAT is another potential PSC constituent at temperatures below ~190 K and transforms irreversibly into β-NAT at higher temperatures (Tizek et al., 2004, and references therein). The presence of NAT in Arctic PSCs was confirmed by Voigt et al. (2000) using balloon-borne in-situ measurements of PSC particles. Large $HNO_3$-containing particles in Arctic PSCs likely consisting of NAT were also reported by Fahey et al. (2001).

Spang and Remedios (2003) and Spang et al. (2005) identified a "peak-like" signature in the spectral region around 820 cm$^{-1}$ in spaceborne infrared PSC observations by CRISTA (Cryogenic Infrared Spectrometers and Telescopes for the Atmosphere) aboard the space shuttle and MIPAS (Michelson Interferometer for Passive Atmospheric Sounding) aboard Envisat. Based on Mie calculations, they suggested that the signature originated from the $\nu_2$ band of the $NO_3^-$ in NAT particles with radii smaller than ~2-3 µm. Höpfner et al. (2006a) attributed the "peak-like" signature around 820 cm$^{-1}$ in the MIPAS-Envisat PSC observations unambiguously to β-NAT by using a new set of refractive indices of β-NAT (Biermann, 1998) and explicit radiative transfer simulations for particles in limb-sounding observations (Höpfner et al., 2002). Particle radii lower than ~3 µm and volume densities higher than ~0.3 µm$^3$/cm$^3$ were derived. To our best knowledge, α-NAT has not been identified in infrared field observations so far.

Further in-situ observations of large potential NAT particles are reported by Brooks et al. (2003) and Molleker et al. (2014) and suggest extremely large particle sizes of 20 µm and more. Particle backward trajectories associated with in-situ observations of large potential NAT particles on 25 January 2010 suggest that the particles had experienced short growth times (Woiwode et al., 2014). However, the sizes of the largest particles derived from the in-situ observations are not reproduced by a state-of-the-art chemistry transport model parameterization (Hoyle et al., 2013; Grooß et al. 2014).

The metastable nitric acid dihydrate (NAD) is another constituent, which is likely present in PSCs and might be involved in the formation of NAT (Worsnop et al., 1993). Nucleation experiments in the cloud chamber AIDA (Aerosol, Interactions, and Dynamics in the Atmosphere) under stratospheric conditions resulted in formation of NAD, while no NAT formation was observed (Stetzer et al., 2006). While there is little evidence for nitric acid hydrates other than NAT in the stratosphere, Kim et al. (2006) reported observations of β-NAT along with NAD as a minor component in the Antarctic stratosphere by the ILAS-II instrument. However, a systematic search in MIPAS-Envisat spectra measured during the Antarctic winter 2003 showed no indication for any NAD particles with radii equal or less than about 1 µm (Höpfner et al., 2006a). As discussed by Grothe et

al. (2004), the spectroscopic data of NAD used in these studies closely corresponds with the α-NAD modification. Furthermore, another metastable high-temperature modification β-NAD has been identified by Grothe et al. (2004) in laboratory experiments at temperatures above ~200 K, which decomposes into β-NAT and NAM (nitric acid monohydrate) at considerably higher temperatures.

We use high resolution MIPAS-STR spectra and collocated in-situ observations during the high-altitude aircraft M-55 *Geophysica* (Stefanutti et al., 1999) ESSenCe (ESA Sounder Campaign 2011, Kaufmann et al., 2013) flight on 11 December 2011 inside an optically thin PSC to study the mid-infrared optical characteristics of large β-NAT particles in detail. The observations are brought into a synoptic context by comparisons with collocated observations by CALIPSO (Cloud-Aerosol Lidar and Infrared Pathfinder Satellite Observation, Pitts et al., 2011, 2013; Lambert et al., 2012, and references therein) and
MIPAS-Envisat (Spang et al., 2005; Höpfner et al., 2006a, 2006b, and references therein).

Using radiative transfer simulations, we model the PSC signatures in the MIPAS-STR observations and constrain the simulations with collocated in-situ particle measurements. To simulate the signatures of spherical to highly aspherical β-NAT particles, we utilize the Mie code by Höpfner et al. (2002, 2006a) and the T-Matrix codes by Mishchenko and Travis (1998) and Bi et al. (2013). The assumption of highly aspherical β-NAT particles is supported by the work of Grothe et al. (2006),
who synthesized highly aspherical β-NAT particles under laboratory conditions and obtained highly aspherical particles with different morphologies depending on the growth conditions.

In Sect. 2, we provide an overview on the meteorological situation, the MIPAS-STR and in-situ PSC observations during the *Geophysica* flight, and collocated CALIPSO and MIPAS-Envisat observations. In Sect. 3, we analyse the PSC signatures in the MIPAS-STR spectra using radiative transfer simulations. In Sect. 4, we compare the particle size distribution supported by
the radiative transfer simulations with the size distributions derived from the in-situ observations, a Chemical Lagrangian Model of the Stratosphere (CLaMS) simulation, and excess $HNO_3$ observed in a nitrification layer below the observed PSC. The results are summarized in Sect. 5.

## 2 PSC observations on 11 December 2011

### 2.1 Instrumentation

MIPAS-STR is a cryogenic Fourier transform spectrometer and provides high resolution limb emission spectra in the mid-infrared (Piesch et al., 1996). Instrument characteristics, data processing, validation and PSC observations by MIPAS-STR are discussed by Woiwode et al. (2012, 2014, 2015). Here, we use MIPAS-STR channel 1 (725 to 990 $cm^{-1}$) and channel 2 (1150 to 1350 $cm^{-1}$) spectra recorded at a spectral sampling of 0.036 $cm^{-1}$ and with an apodized spectral resolution of 0.069 $cm^{-1}$ for cloud detection, retrievals of temperature and trace gases, and the analysis of PSC signatures. The apodized noise equivalent
spectral radiance of the MIPAS-STR spectra is typically ~$10 \cdot 10^{-9}$ W $cm^{-2}$ $sr^{-1}$ cm in channel 1 and ~$8 \cdot 10^{-9}$ W $cm^{-2}$ $sr^{-1}$ cm in channel 2. The vertical spacing of the utilized MIPAS-STR limb observations is mostly 1.0 km (1.5 km for lowest limb views)

between ~5 km and flight altitude. Additional upward sampling provides limited information on the atmospheric scenario above (mainly column information). The horizontal along-track sampling during the discussed flight is ~33 km.

In-situ particle observations aboard the *Geophysica* were performed by the Forward Scattering Spectrometer Probe 100 (FSSP-100) and the Cloud Droplet Probe (CDP) (Molleker et al., 2014, and references therein). Both instruments detect
forward scattering of laser light at 633 nm (FSSP-100) and 658 nm (CDP). From the observations, particle size distributions in the ranges from 1.05 -37 μm (FSSP-100) and 4 -50 μm (CDP) are derived. The FSSP-100 and CDP measurements discussed here were evaluated using Mie theory, and particle sizes are indicated in diameter.

CALIOP (Cloud-Aerosol LIdar with Orthogonal Polarization) aboard the satellite CALIPSO is a two-wavelength polarization-sensitive lidar. CALIOP total and perpendicular backscatter coefficient observations at 532-nm are used for PSC identification
and classification according to Pitts et al. (2011, 2013). CALIPSO is part of the A-train constellation and provides observations up to 82° latitude in each hemisphere. The vertical and horizontal resolution of the PSC product are 180 m and 5 km, respectively.

MIPAS-Envisat is the spaceborne version of the MIPAS instruments (Fischer et al., 2008). Here, we use the spectral window from 780 to 860 cm$^{-1}$ in channel A. The shown observations were performed in the reduced resolution nominal mode and have
an apodized spectral resolution of 0.121 cm$^{-1}$ (von Clarmann et al., 2009). The apodized noise equivalent spectral radiance of the MIPAS-Envisat channel A spectra is typically 11-19·$10^{-9}$ W cm$^{-2}$ sr$^{-1}$ cm. Envisat was deployed on a polar orbit, with the MIPAS-Envisat observations reaching up to ~89°N. The vertical and horizontal sampling of the observations discussed here were ~1.5 km and ~3.5° in latitude (~390 km), respectively.

### 2.2 Meteorological conditions, flight overview and PSC observations aboard the *Geophysica*

Figure 1 gives an overview of the meteorological situation and the observations from the *Geophysica* on 11 December 2011. The flight was performed between 11:05 and 15:00 UTC, with takeoff and landing in Kiruna, Sweden. Also shown are the potential vorticity contours from the ECMWF (European Centre for Medium-Range Weather Forecasts) ERA-Interim reanalysis. The contours indicate that most of the flight was performed inside the polar vortex according to the criterion of Nash et al. (1996). The 430 K potential temperature level corresponds to a geometric altitude of ~18 km, which was the
approximate flight altitude of the *Geophysica*. The flight was performed in a clock-wise pattern, and the polar vortex edge was crossed twice during the flight legs b and c.

Open black circles in Figure 1 indicate the tangent points of the MIPAS-STR limb observations. The horizontal distance of the tangent points from the flight path increases from a few tens of km to ~400 km with decreasing tangent altitude. The cloud index method by Spang et al. (2004) identifies cloud-affected MIPAS observations. In the case of MIPAS-Envisat, cloud index
values higher than 4 are considered cloud-free, and values between 4 and 1 indicate the transition from slightly cloud-affected to opaque conditions. In Figure 1, MIPAS-STR observations with tangent altitudes >16 km (i.e. close to the flight track) and cloud index values <3 are marked by filled circles. Here, we use a less conservative cloud index threshold of 3 to locate the

PSC around the flight track north of ~68°N sharply. The flight section where the FSSP-100 and CDP detected the PSC (see Molleker et al., 2014) is indicated by the red line and coincides well with the MIPAS-STR PSC observations.

Figure 2 shows the vertical flight profile together with the MIPAS-STR tangent points colour-coded with the cloud index. After ascent, the *Geophysica* performed flight legs a and b at an approximately constant flight altitude of ~17 km. In leg c, an approximately constant flight altitude of ~18 km was maintained until ~13:30 UTC. Finally, the *Geophysica* climbed to the ceiling altitude of ~18.5 km, until the descent phase was entered at ~14:20 UTC in leg e. From ~13:30 to 14:20 UTC, low cloud index values indicate the presence of a PSC around flight altitude. Particularly low cloud index values are found in the time interval where the in-situ instruments detected the PSC.

Several of the MIPAS-STR limb scans during the PSC encounter show low cloud index values from flight altitude towards lower stratospheric altitudes. However, no sharp lower boundary of the PSC can be assigned, since the lower limb views are already affected by the PSC around flight altitude. At tropospheric altitudes, the cloud index values decrease towards 1 between 4 and 10 km and indicate tropospheric clouds.

Molleker et al. (2014) calculated backward trajectories for large particles sampled during the discussed flight and found temperatures close to the frost point ~20 h before the flight. While the model temperatures were too warm for ice nucleation, ice particles might have nucleated during lee-wave-induced cooling above Greenland not resolved by the model. Therefore, it is unclear whether the observed particles have nucleated heterogeneously from ice and/or according to a different mechanism.

### 2.3 CALIPSO PSC observations

Figure 3 shows the results of the closest-matching CALIPSO observations (see Fig. 1, 11:23 to 11:26 UTC). The observations were performed ~2 h prior to the *Geophysica* PSC encounter. The scattering ratio at 532 nm is the ratio of total volume backscatter to molecular backscatter and is sensitive to all types of cloud particles (Fig. 3a). The perpendicular backscatter coefficient at 532 nm is sensitive to depolarization of the backscattered light by aspherical cloud particles (Fig. 3b). The combination of inverse scattering ratio and the perpendicular backscatter at 532 nm is utilized for the CALIPSO PSC detection and classification according to Pitts et al. (2011, 2013) (Fig 3c). The method classifies mixtures dominated by STS (STS mix), mixtures of NAT and STS with increasing number densities of NAT (Mix1, Mix2 and Mix2-enh), ice-containing mixtures (Ice mix) and wave ice (Wave ice). Thereby, the utilization of the perpendicular backscatter at 532 nm enhances the detection of tenuous PSCs containing low number densities of NAT particles.

Usually, only CALIPSO nighttime observations are used for PSC detection and classification. Daytime CALIPSO observations are characterized by strongly enhanced noise, making it much more difficult to detect enhanced backscatter from tenuous PSCs. In the case being discussed, strict exclusion of daytime observations would mean excluding CALIPSO observations south of 72.15°N and would greatly decrease the number of close-matching observations. To increase the number of close-matching CALIPSO observations, we included twilight observations down to 67°N. The comparison of the nighttime section >72.15°N and the twilight sections in the cross-sections in Figure 3 however provides a consistent picture of an extended PSC

stretching from ~69°N northwards. False positive signal enhancements due to noise during twilight can be clearly distinguished as patchy spots at latitudes south of ~69°N.

In the vicinity of the *Geophysica* PSC observations, the scattering ratio and the perpendicular backscatter (Fig. 3a and 3b) faintly indicate a PSC between ~19 to 23 km. Between 69.0 and 71.4°N, the cluster of CALIPSO data with low scattering ratios and perpendicular backscatter values slightly exceeding the noise level suggest PSC layers consisting of NAT-containing mixtures (Fig. 3c). Much stronger signal enhancements and a more diverse composition are evident in the part of the PSC extending northward, which spans altitudes from ~18 to 25 km.

## 2.4 PSC signatures in airborne and spaceborne MIPAS observations

Figures 4a and 4b show the MIPAS-STR channel 1 and 2 spectra associated with the uppermost limb views of the limb scans during the time interval where the in-situ instruments detected the PSC (13:39 to 14:19 UTC). Within limb scans at the bottom of a PSC, the views in approximately horizontal geometry typically exhibit the strongest PSC signals. The observation at 13:47 UTC (black) belongs to the limb scan marked in magenta in Figures 1 and 2. A cloud-free observation at 12:27 UTC (grey) is shown for comparison. Compared to the cloud-free observation, the PSC spectra show a strong continuum-like offset of typically 150 to 300 · $10^{-9}$ W cm$^{-2}$ sr$^{-1}$ cm in channel 1 (Fig. 4a). The continuum-like offset results from upwelling greybody-like emission from the surface and/or tropospheric clouds, which is scattered by the PSC particles. A characteristic "shoulder-like" signature is found in the spectral region around 820 cm$^{-1}$, which is shaded in light cyan here and in the following.

The channel 2 spectra (Fig 4b) are characterized by lower overall radiances when compared to channel 1. The spectra show a characteristic continuum-like offset of up to ~150 · $10^{-9}$ W cm$^{-2}$ sr$^{-1}$ cm at the lower boundary of the channel, which decreases towards higher wavenumbers. Absorption signatures of gaseous $H_2O$ are found between 1170 and 1250 cm$^{-1}$ (blue arrows in Fig. 4b) and are the consequence of absorption of upwelling broad-band radiation from the surface and/or low clouds by the abundant $H_2O$ molecules in the lower troposphere (see Höpfner et al., 2002).

Figure 5 shows a spectral window from 780 to 860 cm$^{-1}$ of MIPAS-Envisat channel A observations in the vicinity of the *Geophysica* flight exhibiting PSC signatures (from visual inspection) (see Fig. 1). The observations with tangent altitudes close to the *Geophysica* flight altitude also show a "shoulder-like" signature in the spectral region around 820 cm$^{-1}$ similar to the MIPAS-STR observations.

## 2.5 Ambient conditions and indications for PSC composition

Figure 6 shows the vertical cross-sections of temperature and $HNO_{3(g)}$ retrieved from the MIPAS-STR observations. In Appendix A (Fig. 17) we furthermore show the associated vertical cross-sections of $H_2O_{(g)}$ and $O_3$, since these results are used below. Where applicable, the physical state of $HNO_3$ and $H_2O$ is indicated by "(g)" for "gaseous" and "(s)" for "solid". All other discussed compounds are gaseous under stratospheric conditions, and the physical state is not indicated explicitly.

The vertical cross-section of temperature in Figure 6a typically shows temperatures above 200 K around flight altitude until ~13:00 UTC. From the second half of section c to section e and in particular during the PSC encounter, lower temperatures

between 195-200 K are found above 16 km. The observed temperatures are close to typical values of $T_{NAT}$ (Hanson and Mauersberger, 1988) and above the existence temperatures of STS ($T_{STS}$, ~3.5 K below $T_{NAT}$, Drdla et al., 2003) and ice ($T_{ice}$, ~7 K below $T_{NAT}$, Murphy and Koop, 2005).

The vertical cross-section of $HNO_{3(g)}$ shows mixing ratios increasing with altitude in general (Fig. 6b). In horizontal direction,
the mixing ratios around and below flight altitude slightly decrease in section b and increase again at the beginning of section c, indicating the crossing of the vortex edge (compare Fig. 1). Between 13:15 and 14:15 UTC, a local $HNO_{3(g)}$ maximum peaking at ~16 km is observed. The maximum is located below the PSC encounter of the *Geophysica* (see Sect. 2.1) and extends into the PSC observed at flight altitude. In contrast, the vertical cross-section of the stratospheric tracer $O_3$ in Appendix A (Fig. 17) shows no corresponding maximum, suggesting that the $HNO_{3(g)}$ maximum originated from nitrification.
In Figure 7, we show $T_{NAT}$ calculated from the MIPAS-STR $HNO_{3(g)}$ and $H_2O_{(g)}$ observations during the PSC encounter. $T_{NAT}$ is calculated using the thermodynamic relationships of Hanson and Mauersberger (1988). The $T_{NAT}$ profiles are compared with the corresponding temperature profiles derived from the MIPAS-STR observations, the *Geophysica* UCSE (Unit for Connection with Scientific Equipment) onboard temperature sensor measurements, radiosonde temperature profiles from Bodø and Bjornoya (12Z launches, see Fig. 1) and ECMWF temperature profiles (here: ECMWF T106 grid-point analysis). The
same ECMWF profiles served as the initial guess and a priori profiles for the MIPAS-STR temperature retrievals.

The MIPAS-STR $HNO_{3(g)}$ and $H_2O_{(g)}$ profiles used for the calculation of $T_{NAT}$ are shown in Appendix A (Fig 18). The $H_2O_{(g)}$ profiles are furthermore compared with in-situ observations of total $H_2O$ by FISH (Fast In-situ Stratospheric Hygrometer; Zöger et al., 1999; Meyer et al., 2015) during the same flight for quality assessment.

During the PSC encounter, both MIPAS-STR and the *Geophysica* UCSE in-situ temperature sensor indicate temperatures
between 192-200 K around flight altitude. The UCSE temperatures are slightly lower than the temperatures measured by MIPAS-STR. Both observations are consistent with the radiosonde profiles from Bodø (south of PSC encounter) and Bjornoya (north of PSC encounter). The ECMWF temperatures are in close agreement with the UCSE data. Slightly higher and more variable temperatures are derived from MIPAS-STR around flight altitude and hint at enhanced uncertainties (e.g. additional errors due to a weak cloud index threshold of 2 applied in the retrievals) and/or horizontal temperature variations around the
flight track.

While several MIPAS-STR temperature profiles approach $T_{NAT}$ values already at altitudes >16 km, the UCSE in-situ and ECMWF temperatures approach $T_{NAT}$ at ~18 km. The same $T_{NAT}$ profiles were also calculated considering offsets of (i) +20% in both $HNO_{3(g)}$ and $H_2O_{(g)}$ and (ii) -20 % in both $HNO_{3(g)}$ and $H_2O_{(g)}$ (i.e. offsets for $HNO_{3(g)}$ and $H_2O_{(g)}$ acting into the same direction in terms of $T_{NAT}$). The offset $T_{NAT}$ profiles indicate a weak sensitivity of $T_{NAT}$ to uncertainties in $HNO_{3(g)}$ and $H_2O_{(g)}$.
The close agreement of the temperatures measured by MIPAS-STR, the UCSE data and the ECMWF data with calculated $T_{NAT}$ around flight altitude suggests that the observed PSC was composed of β-NAT. NAD is another possible candidate due to similar formation and persistence conditions when compared with β-NAT (Worsnop et al., 1993; Tizek et al., 2004). α-NAT, STS and ice are unlikely candidates due to too high temperatures.

To investigate the accumulated vertical HNO$_3$ redistribution at the locations inside/below the probed PSC, the correlations of the MIPAS-STR HNO$_{3(g)}$ and O$_3$ profiles associated with the PSC encounter (see Fig. 18a and 18c) and profiles outside the polar vortex are shown in Figure 8. O$_3$ is well suited as stratospheric tracer in this case, since the altitude range under consideration is hardly affected by Arctic winter O$_3$ depletion at this early stage of the polar winter. Therefore, deviations of the correlation inside the polar vortex and inside/below the PSC (green data points) from the unperturbed correlation outside the polar vortex (blue data points) indicate excess HNO$_{3(g)}$ or HNO$_{3(g)}$ deficit due to condensation, sedimentation and evaporation of HNO$_3$-containing particles. The comparison of the correlations shows that inside and below the PSC the retrieved HNO$_{3(g)}$ mixing ratios match or exceed the correlation outside the polar vortex. Maximum differences in HNO$_{3(g)}$ of ~5 ppbv are found for O$_3$ mixing ratios around 1.4 to 1.8 ppmv. For the profiles associated with the PSC encounter, this coincides with the altitude range around the nitrification peak at ~16 km (Fig. 6b). Figure 8 furthermore shows that the vortex correlation around flight altitude and inside the PSC (magenta data points) matches or slightly exceeds the extra-vortex correlation extrapolated to higher altitudes (cyan dashed line). This suggests that any HNO$_{3(s)}$ condensed in the PSC particles represents excess HNO$_3$ originating from particle sedimentation from higher altitudes. Denitrified conditions (i.e. data points on the left side of the extra-vortex correlation) are not found in the altitude range covered by the MIPAS-STR observations.

## 3 Analysis of PSC signatures in the MIPAS-STR spectra

### 3.1 Radiative transfer model and simulation scenario

To simulate the MIPAS-STR spectra, we use the Karlsruhe Optimized and Precise Radiative Transfer Algorithm (KOPRA, Stiller et al., 2002). KOPRA is a fast line-by-line code and allows the modelling of infrared limb observations, including trace gas signatures, cloud parameters and instrumental characteristics. Modelling of MIPAS PSC observations using a Mie model and based on single scattering is discussed and validated by Höpfner et al. (2002, 2006a), Höpfner (2004) and Höpfner and Emde (2004), respectively. In the following, we consider the MIPAS-STR limb view at 13:47 UTC in approximately horizontal orientation (black spectra in Fig. 4a and 4b), which was performed well within the PSC and shows developed PSC signatures. Table 1 provides an overview of the radiative transfer scenarios discussed in the following. For simulating the signatures of β-NAT, we use the "NATcoa" refractive indices by Biermann (1998), which have been applied successfully for the identification for small β-NAT particles by Höpfner et al. (2006a, 2006b). Hereafter, we refer with β-NAT to the "NATcoa" refractive indices by Biermann et al (1998). For simulating the signatures of α-NAT, we use the refractive indices for α-NAT aerosols by Richwine et al. (1995), which show a more developed signature around 820 cm$^{-1}$ than the corresponding faint signature in the refractive indices for α-NAT by Toon et al. (1994). The signatures of NAD are simulated using the refractive indices by Niedzela et al. (1998), which closely correspond with spectroscopic data of α-NAD (Grothe et al., 2004). For simulating STS, we use the refractive indices by Biermann et al. (2000) for 45 wt% solutions of HNO$_3$/H$_2$O, since these compounds are expected to dominate the optical properties of STS (Höpfner et al., 2006a). For ice, we use the refractive indices by Zasetsky et al. (2005) for 200 K.

Particle size distributions are assumed lognormal in shape and characterized by the mode radii (r), geometric standard deviations (i.e. mode width, σ) and mode number densities (N). For simplification, we assume that these quantities are constant over the entire vertical range covered by the PSC. We constrain the used size distributions with the FSSP-100 and CDP in-situ observations during the same flight. In Figure 9a, the particle size distributions derived by Molleker et al. (2014) from the FSSP-100 and CDP observations (black and magenta, respectively) during the PSC encounter are shown. Figure 9b shows the lognormal size distributions used in the radiative transfer simulations of the MIPAS-STR observations (for parameters see Table 2). The bimodal size distribution A (Fig. 9b, red) is adjusted manually to match the size distribution derived from the FSSP-100 observations in terms of shape and condensed $HNO_3$.

To improve the agreement of the simulated spectra with the measurements, only the mode radii of size distribution A are scaled. Accordingly, the particle number densities and mode widths are maintained, while the particle sizes and thereby the volume of $HNO_{3(s)}$ are scaled. In Figure 9b, size distribution B corresponds with the size distribution resulting in the best agreement of the radiative transfer simulations with the MIPAS-STR observations (see below). Size distribution B is obtained from size distribution A by scaling the radii of both modes by a factor of 0.8. Scenarios B1 and B2 are sensitivity calculations with respect to the importance of the larger mode and a hypothetic small particle mode with enhanced number density.

For modelling of aspherical particles, spheroids are assumed. The aspect ratio (AR) is the ratio of the horizontal and rotational semi-axis. Elongated spheroids are characterized by AR < 1, and oblate spheroids by AR > 1. For easy comparisons, the sizes of aspherical particles are given in r or D of volume-equivalent spherical particles.

In the simulations, we adopt a vertical PSC extent from 17-23 km altitude, which is supported by the MIPAS-STR and CALIPSO observations (see Sect. 2.2 and 2.3). The tropospheric cloud scenario is defined considering the MIPAS-STR cloud index values of the associated limb scan. Further information on the tropospheric cloud scenario is inferred from the fact that tropospheric $H_2O_{(g)}$ absorption signatures are identified in the MIPAS-STR spectra (see Fig. 4b). While the MIPAS-STR cloud index values suggest a tropospheric cloud top slightly below 10 km (see Fig. 2), the presence of the $H_2O_{(g)}$ absorption lines requires partially transparent conditions down to low tropospheric altitudes (Höpfner et al., 2002). Both criteria are fulfilled by assuming a partially transparent tropospheric cloud from 0 to 10 km characterized by a low continuum extinction coefficient α of 0.0223 km$^{-1}$ (i.e. ~98 % transmission per 1 km-layer), providing efficient transmission of radiation from nadir directions and increasing opaqueness towards limb geometry at altitudes covered by the tropospheric clouds. Note however that this is a simplified assumption for a probably much more complex and horizontally variable cloud scenario in the reality.

A sea surface temperature of ~280.2 K is adopted as surface temperature from the NOAA Optimum Interpolation (OI) SST V2 GraDS image of the flight day (see http://www.esrl.noaa.gov/psd/) in combination with a sea surface emissivity of 0.99 (see Newman et al., 2005).

Vertical profiles of trace gases required for the simulations of the spectra are retrieved previously from the full cloud-filtered MIPAS-STR limb scans. The retrievals of temperature (utilizing $CO_2$ signatures), $O_3$, $HNO_{3(g)}$, $ClONO_2$, CFC-11, and CFC-12 are performed according to Woiwode et al. (2012, 2015). $H_2O_{(g)}$, $N_2O$, $CH_4$, $COF_2$, CFC-22 and $CCl_4$ are retrieved from the MIPAS-STR channel 1 and 2 spectra in specific microwindows using the same strategy. As the MIPAS-STR limb

observations provide only information above the tropospheric cloud top, the profiles are extrapolated to lower altitudes based on the slopes of the a priori profiles. The profiles of temperature and the strong tropospheric absorber $H_2O_{(g)}$ below the tropospheric cloud top are constructed from the radiosonde observations from Bodø and Bjornoya (see Fig. 1). For $CO_2$, the corresponding polar winter profile for MIPAS of Remedios et al. (2007) is updated for the polar winter 2011/12. For the minor

constituents $SF_6$, CFC-14 and $N_2O_5$, the polar winter profiles from Remedios et al. (2007) are adopted.

### 3.2 Mie calculations of spherical β-NAT, α-NAT, NAD, STS and ice particles

For simulations of the ensemble-averaged single scattering parameters of spherical β-NAT, α-NAT, NAD, STS and ice particles, we use the Mie model coupled to KOPRA (Höpfner et al., 2002, 2006a; Höpfner, 2004). The Mie model provides the wavenumber-dependent ensemble-averaged phase functions, extinction cross-sections and single scattering albedos

required for the simulation of the scattering and absorption/emission of radiation by the particles.

Figure 10 shows the results of the Mie calculations based on size distributions A and B. The upper panels show the simulated spectra together with the observed MIPAS-STR channel 1 and 2 spectra. The residuals between simulation and observation are shown in the lower panels. For size distribution A, the Mie calculations of β-NAT show systematically higher radiances in channel 1 by typically about 50 to $100 \cdot 10^{-9}$ W cm$^{-2}$ sr$^{-1}$ cm when compared to the measurement (Fig. 10a and 10b, red). The

residual values decrease towards the lower and upper boundary of the channel. A characteristic dip is found in the spectral region around 820 cm$^{-1}$, where the "shoulder-like" signature is located in the observations.

Enhanced "line-like" residuals in the densely populated spectral regions below 810 cm$^{-1}$ (mostly $CO_2$ and $O_3$), between 850 to 920 cm$^{-1}$ ($HNO_{3(g)}$), as well as above 970 cm$^{-1}$ ($O_3$), are explained by uncertainties in the knowledge of the profiles of the corresponding trace gases at tropospheric altitudes and above the flight path. The uncertainties of these parts of the profiles

are translated into the simulated spectra by the simulated scattering of radiation by the particles from outside into the field-of-view. Further sources of uncertainties are limitations in the simulated scattering scenario (e.g. limited number of scattering angles). Furthermore, the noise equivalent spectral radiance of the MIPAS-STR spectra steeply increases towards the channel boundaries and enhances the variability in the residuals in these regions.

In channel 2, the simulated radiances are systematically lower by $\sim 25 \cdot 10^{-9}$ W cm$^{-2}$ sr$^{-1}$ cm below 1300 cm$^{-1}$ and show better

agreement with the measurements at higher wavenumbers for size distribution A. Similar to the observation, prominent $H_2O_{(g)}$ absorption signatures are found between 1170 and 1250 cm$^{-1}$ in the simulation (indicated black arrows in Fig. 10 to Fig. 12). Further "line-like" residuals exceeding the noise level are again explained by uncertainties in the knowledge of the profiles of the corresponding trace gases at tropospheric altitudes and above the flight path (here mostly $O_3$, $N_2O$, $CH_4$ and $HNO_3$), limitations in the simulated scattering scenario and increasing noise towards the channel boundaries.

The results for size distribution B (Fig. 10a and 10b, blue) show improved agreement in channel 1. However, a dip is observed again in the spectral region around 820 cm$^{-1}$, and the residual values are increasingly negative towards the lower and upper boundary of channel 1. In channel 2, the residual values below 1300 cm$^{-1}$ are more negative when compared to size distribution A.

For α-NAT (Fig. 10c and 10d), NAD (Fig. 10e and 10f), and STS (Fig. 10g and 10h), similar residuals are found for both size distributions when compared to β-NAT. For these species, more "step-like" residual signatures are found in the spectral region around 820 cm$^{-1}$ for both size distributions when compared to β-NAT. When compared with the previous cases, the simulation of STS using size distribution A results in the best agreement with the observations in channel 2 (Fig. 1h, red).

For ice (Fig. 10i and 10j), the worst overall agreement is found in channel 1. For both size distributions, the simulated spectrum shows a negative overall gradient from the lower to the upper boundary of the channel. Negative residual values as low as -200·10$^{-9}$ W cm$^{-2}$ sr$^{-1}$ cm are found at the upper end of channel 1 for both size distributions, and again a "step-like" dip is found in the spectral region around 820 cm$^{-1}$. In channel 2, the observed radiances are slightly overestimated below ~1250 cm$^{-1}$ for size distribution A and significantly underestimated below 1300 cm$^{-1}$ for size distribution B.

Overall, the results of the Mie simulations show worse agreement with the observations and prominent residual signatures in the spectral region around 820 cm$^{-1}$. In the following simulations of aspherical particles, we exclude α-NAT, NAD, STS and ice as dominating constituents of the observed PSC for the following reasons:

- β-NAT exhibits a characteristic spectral signature around 820 cm$^{-1}$ and is the only PSC constituent known to be thermodynamically stable under the conditions of the flight.

- The metastable α-NAT modification also exhibits a characteristic spectral signature around 820 cm$^{-1}$ with a weaker amplitude (Höpfner et al. 2006a). However, α-NAT is expected to irreversibly transform into β-NAT at the temperatures around flight altitude (Tizek et al., 2004; compare Fig. 7). Furthermore, laboratory experiments by Grothe et al. (2006) support approximately spherical geometries of α-NAT particles, while the presented Mie calculations for spherical α-NAT particles result only in coarse agreement with the observations.

- While α-NAD shows a similar spectral signature with weaker amplitude centred at 808 cm$^{-1}$ (Niedziela et al., 1998, Grothe et al., 2004), the signature is not capable of reproducing the residual dip slightly below 820 cm$^{-1}$ (see Sect. 3.5). The same is expected for the high-temperature modification β-NAD, which was characterized by Grothe et al. (2004) in laboratory experiments and shows a similar spectral signature centred at 811 cm$^{-1}$. Furthermore, the observations indicate temperatures close to the threshold temperature of β-NAT and slightly too warm for NAD under
stratospheric conditions.

- STS droplets are not expected to grow to the large sizes observed in-situ and are not supported by the observed temperatures (e.g. Peter, 1998). Furthermore, STS droplets are expected to be approximately spherical. However, the presented Mie calculations for spherical STS droplets result only in coarse agreement with the observations.

- For ice, the observed temperatures are considerably too warm, and the Mie calculations result in the worst overall
agreement.

### 3.3 T-Matrix calculations of β-NAT particles with moderate aspect ratios

For simulations of the ensemble-averaged single scattering parameters of moderately aspherical β-NAT particles, we use the double-precision T-Matrix code for polydisperse, randomly oriented, rotationally symmetric particles by Mishchenko and Travis (1998). The T-Matrix is calculated according to the extended boundary condition method (EBCM) (Waterman, 1971). Advantages of this code are fast computations for moderately aspherical particles and the capability of direct coupling to KOPRA. The same setup as in the Mie calculations is used.

The results for elongated spheroids with AR=0.5 and oblate spheroids with AR=2.0 are shown in Figure 11. Briefly, the results for both AR values are very similar to each other and to the Mie scenario (compare Fig. 10a and b) for both size distributions. Again, a characteristic dip is found in the spectral region around 820 cm$^{-1}$ for both AR values and size distributions.

### 3.4 T-Matrix calculations of β-NAT particles with extreme aspect ratios

For simulations of the ensemble-averaged single scattering parameters of highly aspherical β-NAT particles, we use invariant imbedding and separation of variables (IIM+SOV) T-Matrix calculations (Bi et al., 2013). Again, polydisperse, randomly oriented spheroids and the same setup as in the previous scenarios are considered. To allow flexible variation of size distributions and save computing time, the optical parameters of β-NAT particles are calculated for discrete wavenumbers ν (Δν=1 cm$^{-1}$) and sizes (r=0.5 to 20 µm with Δr=0.5 µm) and are tabulated. Particle size distributions are implemented by means of binned lognormal size distributions referring to the tabulated optical parameters. The choice of the AR values is motivated by the laboratory experiments by Grothe et al. (2006), resulting in needles and platelets with similar proportions depending on the crystallization conditions.

The results of the simulations with AR=0.1 and AR=10.0 are shown in Figure 12. For AR=0.1 and size distribution A, high systematic offsets of 100 to 200·10$^{-9}$ W cm$^{-2}$ sr$^{-1}$ cm are found in channel 1 and up to ~50·10$^{-9}$ W cm$^{-2}$ sr$^{-1}$ cm in channel 2 (Fig. 12a and 12b, red). In contrast, size distribution B results in a much flatter residual pattern close to zero in wide ranges of channel 1 and in the entire channel 2 (Fig. 12a and 12b, blue). For both AR values and size distributions, the residual peak in the spectral region around 820 cm$^{-1}$ flattens out to a high degree in contrast to the previous simulations. The H$_2$O$_{(g)}$ absorption signatures in the spectral range from 1170 to 1250 cm$^{-1}$ are reproduced well for size distribution B.

For AR=10.0 and size distribution A, again high residual values of typically around 100 to 150·10$^{-9}$ W cm$^{-2}$ sr$^{-1}$ cm are found in channel 1 and mostly below 50·10$^{-9}$ W cm$^{-2}$ sr$^{-1}$ cm in channel 2 (Fig. 12c and 12d, red), whereas size distribution B results in improved agreement with the observation (Fig. 12c and 12d, blue). When compared to the corresponding AR=0.1 scenario, increasingly negative residual values are found towards the upper boundary of channel 1, and slightly negative residual values below 1300 cm$^{-1}$ in channel 2. However, the spectral region around 820 cm$^{-1}$ in channel 1 is reproduced slightly better than in the AR=0.1 case.

Both scenarios, elongated spheroids with AR=0.1 and oblate spheroids with AR=10.0 result in considerably improved agreement with the observations when compared with the previous simulations. For both size distributions, the residual dip in the spectral region around 820 cm$^{-1}$ flattens out significantly.

For size distribution B, the spectral range above 900 cm$^{-1}$ is furthermore reproduced well by the AR=0.1 scenario, and good agreement is found in channel 2. As in the previous cases, residual patterns remain especially in the densely populated spectral regions and at the channel boundaries.

### 3.5 Sensitivity to aspect ratio and comparison with refractive indices

In the following, we compare the radiative transfer simulations of elongated spheroids consisting of β-NAT with different aspect ratios. The same comparison for oblate spheroids results in similar conclusions. Figures 13a and 13b show the results of the Mie scenario (AR=1.0, cyan) and the T-Matrix calculations with AR=0.5 (magenta) and AR=0.1 (green) for size distribution B.

The scenarios with AR=1.0 and 0.5 show only small differences of a few 10$^{-9}$ W cm$^{-2}$ sr$^{-1}$ cm. For AR=0.5, the residual peak in the spectral region around 820 cm$^{-1}$ decreases slightly, and the spectrum is tilted slightly towards the observation in the higher wavenumber region of channel 1. Slightly improved agreement with the observation is found also in channel 2. In contrast, the agreement with the observation is improved considerably for AR=0.1: in channel 1, the residual dip in the spectral region around 820 cm$^{-1}$ flattens out to a high degree. The simulated spectrum is tilted towards the observation above 900 cm$^{-1}$ and reproduces the observed spectrum in this region well. In channel 2, higher radiances in close agreement with the observation are found when compared to the other scenarios.

Figure 13c and 13d show the imaginary parts of the refractive indices of β-NAT, α-NAT, NAD, STS and ice, which determine the absorption and emission characteristics of the particles. In the spectral region around 820 cm$^{-1}$, which is weakly populated by trace gas signatures, β-NAT provides the strongest signature due to the $\nu_2$ symmetric deformation mode of NO$_3^-$. Further broad peaks are found in channel 2 around ~1330 cm$^{-1}$ due to the $\nu_3$ asymmetric stretch mode of NO$_3^-$ and around ~1200 cm$^{-1}$ due to the $\nu_2$ symmetric umbrella mode of H$_3$O$^+$ (Ortega et al., 2006, and references therein). These signatures however result in weak and broad residual patterns in the Mie scenarios, which are unspecific without further information (see Fig.10). The α-NAT refractive index imaginary part also shows a peak at 820 cm$^{-1}$ due to the $\nu_2$ symmetric deformation mode of NO$_3^-$ of this NAT modification. The amplitude is however only about half of the amplitude of the used β-NAT refractive index, and it is superimposed by a broader signature beginning at ~900 cm$^{-1}$ and peaking at ~780 cm$^{-1}$. Similarly to α-NAT, NAD also shows a peak at 808 cm$^{-1}$ with a smaller amplitude when compared to β-NAT along with another peak with higher amplitude at 745 cm$^{-1}$ in the region densely populated with trace gas signatures. Furthermore, NAD also shows broad peaks at ~1260 cm$^{-1}$ due to the $\nu_3$ mode NO$_3^-$ and at ~1150 cm$^{-1}$ (coinciding with the channel boundary) due to the $\nu_2$ mode of H$_3$O$^+$. For α-NAT, only the increasing slopes of the corresponding signatures towards the boundaries of channel 2 can be identified. STS shows no significant peak-like signatures in both channels. For ice, a broad peak spanning the entire range of channel 1 is found due to librational modes in the ice crystal (Zasetsky et al., 2005).

Figures 13e and 13f show the real parts of the refractive indices of the discussed species, which determine the scattering of radiation from outside into the field-of-view. β-NAT shows the strongest "step-like" signature in the spectral region around 820 cm$^{-1}$, corresponding with the peak in the refractive index imaginary part. α-NAT and NAD again show weaker corresponding signatures at 820 cm$^{-1}$ and 808 cm$^{-1}$. For NAD, another sharp step is seen at ~745 cm$^{-1}$ and a broad step around ~1270 cm$^{-1}$. Ice shows an extended decreasing gradient from the lower boundary of channel 1 to ~930 cm$^{-1}$. Neither ice, nor STS shows any characteristic sharp signature in both channels.

The refractive indices of β-NAT show the strongest signature at 820 cm$^{-1}$ in both the refractive index imaginary and real part and allow to model the "shoulder-like" signature around 820 cm$^{-1}$ with a realistic spectral position and amplitude. α-NAT and NAD also show similar but significantly weaker signatures at 820 cm$^{-1}$ and 808 cm$^{-1}$, respectively. Therefore, potential highly aspherical large α-NAT and NAD particles are expected to result in considerably weaker corresponding "shoulder-like" signatures along with further discrepancies from β-NAT as a consequence of different patterns in the refractive index imaginary and real parts.

Figure 13g and 13h show the ensemble-averaged absorption and scattering cross-sections of β-NAT for the considered size distribution for AR=0.1 and AR=1.0, which determine the absorption/emission and scattering characteristics of the simulated particles. The T-Matrix scenario with AR=0.1 shows a much stronger peak in the absorption cross-section and a stronger step in the scattering cross-section in the spectral window around 820 cm$^{-1}$ when compared to the Mie scenario, which together result in the characteristic "shoulder-like" signature in the simulated spectrum. Furthermore, the AR=0.1 scenario shows considerably higher values of the scattering cross-section towards higher wavenumbers, resulting in a relatively flat baseline of the simulated spectrum towards the upper end of channel 1. In the AR=0.1 scenario, higher absorption and scattering cross-sections in channel 2 result in higher radiances in the corresponding simulated spectrum.

The relative contributions of absorption/emission and scattering to the observed signature is investigated in Figure 14 for the AR=1.0 scenario (magenta) and the AR=0.1 (green) scenario. Also shown are the same simulations neglecting the scattering by the particles (violet and dark green, respectively). Thereby, the wavenumber-dependent ensemble-averaged particle extinction coefficients are calculated taking into account only absorption and emission by the particles and neglecting the scattering source function.

For AR=0.1, the scenario neglecting scattering by the particles results in lower radiances by up to ~200·10$^{-9}$ W cm$^{-2}$ sr$^{-1}$ cm in channel 1 (Fig. 14a) and ~100·10$^{-9}$ W cm$^{-2}$ sr$^{-1}$ cm in channel 2 (Fig. 14b) when compared to the same scenario including scattering. This demonstrates the importance of the scattering source function. The same effect is found to a lesser degree also in the corresponding AR=1.0 scenarios. The AR=1.0 and the AR=0.1 scenarios neglecting scattering show only small differences in the order of 10·10$^{-9}$ W cm$^{-2}$ sr$^{-1}$ cm from each other. Enhanced differences of ~30·10$^{-9}$ W cm$^{-2}$ sr$^{-1}$ cm are however found in the spectral region around 820 cm$^{-1}$, where a weak "peak-like" signature with a maximum at ~817 cm$^{-1}$ can be identified in the AR=0.1 scenario due to the net emission by the 820 cm$^{-1}$ mode of β-NAT.

Much higher discrepancies between the AR=1.0 and AR=0.1 scenario are seen if scattering is included, and a "shoulder-like" net signature peaking at ~814 cm$^{-1}$ is found in the AR=0.1 scenario. In the observations, the shoulder-like signature is more

narrow than in the simulation and peaks at ~818 cm$^{-1}$. The simulation represents the increasing flank from higher towards lower radiances very well, while the maximum and the decreasing flank towards lower radiances are located at lower wavenumbers and are broader than in the simulation.

We point out that there are still remaining systematic discrepancies between the simulation of β-NAT for AR=0.1 and the observation. Beside the limitations of the radiative transfer simulation and scenario, the discrepancies are attributed to uncertainties in the refractive indices used (30% uncertainty for imaginary parts, see Höpfner et al., 2006a; not quantified for real parts) and the fact that the refractive indices are only available at a spectral resolution of 1 cm$^{-1}$. In contrast, the MIPAS-STR observations have a considerably higher apodized resolution of 0.069 cm$^{-1}$. Another reason for the observed discrepancies might be the simplified particle shape assumed in the simulations. Real β-NAT particles might have different and more complex shapes and are likely to have edges and flat surfaces, altering their optical characteristics. This is supported by the experiments of Grothe et al. (2006), resulting in highly aspherical β-NAT particles (i.e. platelets and needles).

## 3.6 Sensitivity calculations on tropospheric scenario for β-NAT, AR=0.1

In the following, we investigate the sensitivity of the AR=0.1 scenario for β-NAT and size distribution B on modifications of the tropospheric cloud scenario and sea surface temperature. Figure 15 shows this scenario (Fig. 15a-15d, green, hereafter "optimized scenario") together with sensitivity simulations considering modified tropospheric cloud scenarios (see Table 1). The optimized scenario uses a tropospheric cloud layer between 0 and 10 km characterized by a continuum absorption coefficient of 0.0223 km$^{-1}$ (at all wavelengths) and a sea surface temperature of 280.2 K (see Sect. 3.1).

The first modified tropospheric cloud scenario considering an opaque tropospheric cloud from 0 to 2 km and fully transparent conditions above hardly changes the simulated spectra in both channels when compared to the optimized scenario (Fig 15a and 15b, magenta). This modified scenario is motivated by CALIPSO observations of tropospheric clouds during the scan from 11:23 to 11:26 (not shown, see http://www-calipso.larc.nasa.gov/products/), indicating a dense cloud layer below ~2 km. Note however that CALIPSO provides vertical cross-sections corresponding with a narrow footprint and that the horizontal cloud distribution is variable. The second modified scenario neglects the presence of any tropospheric clouds (Fig. 15a and 15b, cyan). The comparison with the optimized scenario and the observation shows that the radiances in both channels are significantly overestimated. However, a "shoulder-like" signature in the spectral region around 820 cm$^{-1}$ can be identified. The third modified scenario considers the extreme case of an opaque tropospheric cloud between 0 to 10 km (Fig. 15c and 15d, cyan). High negative offsets are found in both channels when compared to the observation, and the H$_2$O$_{(g)}$ absorption signatures between 1170 and 1250 cm$^{-1}$ are not reproduced. Accordingly, this scenario confirms the requirement for partially transparent conditions down to low tropospheric altitudes. However, also this scenario weakly indicates a "shoulder-like" signature around 820 cm$^{-1}$. Finally, the scenario with a 7 K lower surface temperature (i.e. 273.2 K, Figs 15c and 15d, magenta) results only in small changes when compared to the optimized scenario. Accordingly, the identification of the "shoulder-like" signature around 820 cm$^{-1}$ is robust against the adopted tropospheric cloud and tropospheric radiation emission scenario. The presence of partially transparent conditions down to the low troposphere is confirmed.

### 3.7 Sensitivity calculations on PSC scenario for β-NAT, AR=0.1

In the following, the sensitivity of the simulated spectra to the thickness of the simulated PSC layer and the adopted particle size distribution is investigated. Figure 16 shows the optimized AR=0.1 scenario (green) using size distribution B together with sensitivity simulations considering a reduced PSC top altitude and modified size distributions (see Fig. 9b and Tables 1 and 2). The scenario with a lower PSC top altitude of 20 km instead of 23 km shows a significant negative offset in both channels (Fig. 16a and 16b, orange). Both, the signature around 820 cm$^{-1}$ in channel 1 and the $H_2O_{(g)}$ absorption signatures in channel 2 show smaller amplitudes when compared to the optimized scenario.

The scenario considering only the second mode of the size distribution B (size distribution B1, see Fig. 9b and Table 2) results only in small changes when compared to the optimized scenario (Fig. 16a and 16b, cyan). Accordingly, the first mode of size distribution B is of minor importance for the simulated radiances. We furthermore investigate the role of a hypothetic mode of small β-NAT particles with a high number density, which might have been present above the flight path and contributed to the observed spectra. In size distribution B2 (see Fig. 9b and Table 2), the first mode is replaced by a small particle mode with a mode radius of 0.5 µm and a number density enhanced by a factor of 100. The corresponding total amount of condensed $HNO_{3(g)}$ is comparable with size distribution B (see Table 3). The resulting simulated spectra (Fig. 16a and 16b, magenta) are almost identical to the optimized scenario. Accordingly, the exact shape of the first mode is of minor importance, as long as the amount of condensed $HNO_{3(g)}$ is maintained.

So in summary, reducing the simulated PSC layer thickness by 50 % results in significantly lower simulated radiances. However, the overall spectral pattern, a weak signature around 820 cm$^{-1}$ and $H_2O_{(g)}$ absorption signatures between 1170 and 1250 cm$^{-1}$ can be identified. Omitting the first mode of particle size distribution B or replacing the first mode by a small particle mode with similar $HNO_3$-content result only in small changes in the simulated spectra. Accordingly, the identification of the β-NAT signature around 820 cm$^{-1}$ is robust against the assumed thickness of the PSC layer and modifications of the size distribution in the considered ranges. The second mode of size distribution B with a mode radius of 4.8 µm plays the key role in the simulation of the overall offset due to scattering and the "shoulder-like" signature around 820 cm$^{-1}$.

We furthermore perform a sensitivity study based on the scenario involving the simplified size distribution B1 to investigate the effect of decreasing mode radii on the observed spectral signatures when the total volume of β-NAT is kept constant. The results are reported in Appendix B and show that the transition from a "shoulder-like" to a "peak-like" signature occurs for AR=0.1 and the considered mode width at a mode radius of ~3.0 µm. For a mode radius of 1.0 µm, a "peak-like" signature is found in agreement with a corresponding Mie simulation. The results show furthermore, that a modified "shoulder-like" signature along with further changes in the simulated spectra results for spherical particles with a mode radius of 3.0 µm.

### 4 Comparison of simulated and observed particle size distributions

In the following, we compare the amounts of gas-phase equivalent $HNO_3$ corresponding with size distributions A, B, B1 and B2 with the same quantity derived from the in-situ observations by the FSSP-100 and CDP (see Tables 2 and 3 and Fig. 9).

Gas-phase equivalent $HNO_3$ is calculated for pressure and temperature at flight altitude and corresponds with the volume mixing ratio of gaseous $HNO_3$ added to the gas-phase if the particles would evaporate instantaneously. The results are furthermore compared with a corresponding size distribution simulated by the CLaMS for the area from 70.0°N to 72.0°N and 15.0°E to 22.5°E in the altitude range from ~18.0 to 18.5 km at 14:00 UTC (Fig. 9b). Finally, the results are compared with

excess $HNO_{3(g)}$ in the nitrification layer below the flight path retrieved from the MIPAS-STR measurements.

The bimodal size distribution A corresponds with 18.2 ppbv gas-phase equivalent $HNO_3$ and resembles approximately the size distribution derived from the FSSP-100 observations, which corresponds to 18.5 ppbv of gas-phase equivalent $HNO_3$. The optimized size distribution B and size distributions B1 and B2 correspond to gas-phase equivalent $HNO_3$ from 8.4 to 9.8 ppbv. Size distribution B was derived from size distribution A by reducing the mode radii of both modes by 20 %. The corresponding

large difference in condensed $HNO_3$ is the consequence of the high sensitivity of particle volume on particle radius.

The CLaMS size distribution shown in Figure 9b includes an STS mode (below 3 µm in the shown range of dN/dlog(D)) and a binned NAT mode. While the small STS mode hardly contributes to the total amount of condensed $HNO_3$ in the simulation, the NAT mode contains a considerably smaller amount of gas-phase equivalent $HNO_3$ of only ~1.6 ppbv when compared to size distribution B and more than one order of magnitude less gas-phase equivalent $HNO_3$ than derived from the in situ

observations. In the CLaMS simulation, a more narrow NAT mode results, peaking at the size bin centred at r=3.7 µm. The largest particles are found in the size bin centred at r=4.7 µm.

While the occurrence of PSCs and overall denitrification are reproduced well by CLaMS in general (Grooß et al., 2014), the discrepancy between the size distribution simulated by CLaMS and derived from the observations discussed here might by linked to uncertainties in the simulated NAT nucleation rates, missing nucleation processes (e.g. NAT nucleation on pre-

existing ice) and simulated particle growth rates.

Furthermore, sedimentation rates for spherical particles are assumed in the CLaMS simulations. However, aspherical particles, such as supported by the observations discussed here, would have lower sedimentation rates than spherical NAT particles, which affects the time-dependent development of particle size distributions inside PSCs and $HNO_3$ redistribution in general (Woiwode et al., 2014). Furthermore, locations and altitudes of specific PSCs and atmospheric structures in model simulations

and observations can differ from each other, while the overall processes are represented in a realistic way in the model simulations.

The maximum amount of excess $HNO_{3(g)}$ of ~5 ppbv in the nitrification layer below the flight track (i.e. exceeding the extra-vortex correlation in Fig. 8) is used to put the amounts of condensed $HNO_{3(s)}$ corresponding with the different size distributions into a perspective. $HNO_{3(s)}$ at flight altitude corresponding with the size distributions also represents excess $HNO_3$ originating

from higher altitudes, as (i) the $O_3$-$HNO_{3(g)}$ correlation at flight altitude already matches or slightly exceeds the extra-vortex correlation and (ii) the large particle sizes derived from the observations imply significant sedimentation rates and therefore an origin from higher altitudes. Converting the amount of excess $HNO_3$ present at ~16 km in term of molecules per volume to the pressure at the flight altitude of ~18.5 km results in a mixing ratio of ~7.5 ppbv, which is comparable with gas-phase equivalent $HNO_3$ derived from size distributions B, B1 and B2. However, this comparison has to be taken with care, since the

excess $HNO_{3(g)}$ in the nitrification layer might have resulted from gradual evaporation of large particles settling into layers with temperatures above $T_{NAT}$ rather than simultaneous evaporation of a certain PSC layer due to raising temperatures. The spatial coincidence of the PSC and the nitrification layer below, the observations of large particles and the fact that temperatures exceed $T_{NAT}$ below the flight path strongly suggest that an ongoing denitrification process was observed and that the nitrification layer below the flight path was associated with the PSC above.

Accordingly, the size distributions B, B1 and B2 supported by the simulations of the MIPAS-STR spectra suggest by factors of ~2 to 3 smaller amounts of condensed $HNO_3$ when compared to the size distributions derived from the in-situ observations. We mention that Borrmann et al. (2000) investigated the effects of spheroids with AR=0.5 on FSSP observations. Similar to the infrared observations discussed here, the results were close to corresponding Mie calculations. However, the effects of highly aspherical particles on the interpretation of FSSP measurements are uncertain and might explain this discrepancy.

The larger particle sizes derived from the FSSP-100 and CDP measurements using the Mie theory are not necessarily in contradiction with the radiative transfer simulations of the MIPAS-STR observations discussed here when interpreted as maximum dimensions of highly aspherical particles. For example, elongated spheroids with extreme aspect ratios can easily span lengths of several tens of microns while having relatively small individual particle volumes. Evidence of particles with sizes of this magnitude is provided by CIP shadow cast images recorded during the Arctic winter 2009/10 (Molleker et al. 2014).

The CLaMS simulation suggests a smaller amount of condensed $HNO_3$ by a factor of ~6 when compared with size distribution B and by more than one order of magnitude less condensed $HNO_3$ when compared with the size distributions derived from the in-situ observations. The amount of excess $HNO_{3(g)}$ in the nitrification layer below the flight path is of the same magnitude as $HNO_{3(s)}$ corresponding with size distributions B, B1 and B2.

**5 Conclusions**

The MIPAS-STR observations associated with the Arctic PSC flight on 11 December 2011 show a characteristic "shoulder-like" signature in the spectral region around 820 cm$^{-1}$, which is attributed to the $\nu_2$ symmetric deformation mode of $NO_3^-$ of β-NAT. The observed signature is explained by the absorption/emission and scattering characteristics of large highly aspheric β-NAT particles. While Mie calculations and T-Matrix calculations assuming spheroids with moderate AR values of 0.5 and 2.0 do not reproduce the observed signature and the overall spectral patterns, T-Matrix calculations for spheroids involving extreme AR values of 0.1 and 10.0 result in reasonable agreement with the observations. In the discussed scenario, best agreement is found for AR=0.1 and a bimodal lognormal size distribution with mode radii of 2.0 and 4.8 µm when adopting a particle size distribution similar in shape with the results from the collocated in-situ observations. The smaller mode with a mode radius of 2.0 µm plays only a minor role in the simulations. The fact that best agreement is found for highly elongated particles might hint on a heterogeneous nucleation of the particles involving ice and a subsequent phase transition from α-NAT to β-NAT (Grothe et al., 2006, Iannarelli et al., 2016, Weiss et al. 2016, and references therein). While the temperatures at

flight altitude were too warm for ice and α-NAT and model temperatures do not support a previous ice nucleation, Molleker et al. (2014) suggest that ice particles might have nucleated previously during lee-wave-induced cooling above Greenland not resolved by the model, enabling an ice-induced nucleation of NAT.

The $\nu_2$ symmetric deformation mode of $NO_3^-$ at 820 cm$^{-1}$ is well suited for identification of β-NAT, since it is located in a

spectral region weakly populated by gaseous absorbers and where the thermal emission of the atmosphere is higher than at higher wavenumbers. Furthermore, it represents a sharp feature, having a significant amplitude in both the refractive index imaginary and real part. The identification of large aspherical β-NAT particles is furthermore supported by the reasonable overall agreement of the simulations with the observations in the entire spectral regions analyzed, covering further weak and broad signals due to the $\nu_3$ asymmetric stretch mode of $NO_3^-$ around ~1330 cm$^{-1}$, the $\nu_2$ symmetric umbrella mode of $H_3O^+$ at

~1200 cm$^{-1}$ and unspecified patterns. The combination of these signatures results in a specific fingerprint in the absorption and scattering cross-sections of highly aspherical large β-NAT particles. Potential highly aspherical large α-NAT and α-NAD particles are expected to result in considerably weaker "shoulder-like" signatures at ~820 cm$^{-1}$ and ~808 cm$^{-1}$ due to weaker corresponding signatures in the refractive indices along with further discrepancies from β-NAT due to different patterns in the refractive indices at higher wavenumbers.

Sensitivity calculations involving a simplified size distribution show that for AR=0.1 the transition from a "shoulder-like" to a "peak-like" signature occurs at a mode radius of ~3.0 µm. A developed "peak-like" signature as discussed by Höpfner et al. (2006a) is found for a mode radius of 1.0 µm, which is almost identical to the corresponding Mie simulation. Furthermore, a corresponding Mie simulation with a mode radius of 3.0 µm shows that a modified "shoulder-like" signature along with further changes in the modelled spectra can be simulated for spherical particles using the discussed size distribution.

The combination of the MIPAS-STR cloud, temperature and trace gas data products, collocated in-situ observations aboard the *Geophysica*, CALIPSO observations and radiosonde observations allow us to define the radiative transfer scenario accurately. The scattering of upwelling tropospheric radiation by the particles plays an important role in the simulation of the overall spectral patterns and in particular of the "shoulder-like" signature around 820 cm$^{-1}$. The importance of scattering is also reflected by the observation of prominent $H_2O_{(g)}$ absorption signatures in the MIPAS-STR limb emission spectra between

1170 to 1250 cm$^{-1}$, implying scattering of radiation from the low troposphere/surface into the field-of-view. Remaining discrepancies between the radiative transfer simulations and the observations are attributed to the limitations of the radiative transfer simulation and scenario, the uncertainties and the comparably coarse spectral resolution of the available refractive index data, and the uncertain particle geometries. Further combinations of particle shapes (e.g. including edges and flat surfaces), aspect ratios and potential shape-dependent orientation effects might further improve the agreement of simulations

and observations.

The radiative transfer simulations of the MIPAS-STR observations suggest ~20 % smaller radii when compared to the size distributions derived from the in-situ observations, resulting in factors of 2-3 smaller amounts of condensed $HNO_3$. A corresponding CLaMS simulation suggests a factor of ~6 less condensed $HNO_3$ and smaller particles when compared to the radiative transfer simulations. This discrepancy might be linked to uncertainties and limitations of the NAT nucleation, particle

growth and particle sedimentation rates (which would be different for highly aspherical particles) simulated by CLaMS and the limited comparison with the localised case study being discussed. Excess $HNO_{3(g)}$ derived from the MIPAS-STR observations in the nitrification layer below the observed PSC is of the same magnitude as $HNO_{3(s)}$ at flight altitude corresponding with the size distribution supported by the radiative transfer simulations.

The discrepancies between the particle size distributions derived from the MIPAS-STR observations and the in situ observations might be due to the fact that spherical particles were assumed in the evaluation of the in situ observations. On the other hand, the particle sizes derived from the in situ observations may be reconciled with the simulations of the MIPAS-STR observations when interpreted as the maximum dimensions of highly aspherical particles. The results of our radiative transfer calculations suggest smaller volumes of individual particles than suggested by non-optical measurements during other PSC

flights (Fahey et al., 2001; Molleker et al., 2014). However, the  particle sizes derived here suggest significant sedimentation rates and associated vertical redistribution of HNO₃. We mention that the shapes of particle size distributions of PSCs depend on the specific growing conditions of the particles. Furthermore, there might be some diversity in the NAT phase in terms of particle sizes and shapes, which is not captured by the assumption of lognormal size distributions and a single particle geometry. Real stratospheric β-NAT particles might have more complex shapes similar to ice (e.g. Libbrecht, 2005), and

hollow NAT shells were proposed as an alternative explanation for large particle sizes derived from in-situ measurements (Molleker et al., 2014; see also Biermann et al., 1998).

The shapes and sizes of large β-NAT particles are important parameters for simulations of denitrification, chemistry and radiation in the polar stratosphere and for the interpretation of infrared limb observations, in-situ particle measurements and lidar observations. The presented results show that large highly aspherical β-NAT particles show a characteristic spectral

signature, which may be exploited for the identification of large β-NAT particles involved in denitrification. Further laboratory measurements of the refractive indices of β-NAT particles with high spectral resolution would help to improve radiative transfer simulations and retrievals from high resolution mid-infrared limb observations. The application of the T-Matrix method for highly aspherical particle shapes would help in the interpretation of in-situ forward scattering measurements of β-NAT particles (compare Borrmann et al., 2000). Further airborne remote sensing and in-situ PSC observations in the Arctic

stratosphere using the combination of optical and non-optical methods would help to better characterize β-NAT particles, and advanced in-situ measurements would allow actual measurement of the particle shapes (e.g. Kaye et al., 2008).

**Appendix A**

The vertical cross-sections of $H_2O_{(g)}$ and O₃ retrieved from the MIPAS-STR observations are shown in Figure 17. In the O₃ cross-section (Fig. 17b), the mixing ratios around and below flight altitude decrease in section b and increase again at the

beginning of section c similar to HNO₃ (see Fig. 6b) and indicate the crossing of the vortex edge (see Fig. 1).

The absence of a local $O_3$ maximum between 13:15 and 14:15 UTC around ~16 km confirms that the $HNO_{3(g)}$ maximum in Figure 6b is the result of a nitrification process. The vertical profiles of $HNO_{3(g)}$, $H_2O$ and $O_3$ used for the calculation of $T_{NAT}$ and the $O_3$-$HNO_{3(g)}$ correlation in Section 2.5 are shown in Figure 18.

### Appendix B

The goal of the sensitivity study discussed in the following is to identify an approximate size threshold for particles with AR=0.1 for the transition from a "peak-like" (compare Höpfner et al., 2006a) to a "shoulder-like" signature in the spectral region around 820 cm$^{-1}$. Corresponding Mie calculations for spherical particles (AR=1.0) are shown for comparison. Starting point for the simulations is the simplified size distribution B1 (1-modal, r=4.8 µm, see Fig. 9a and Table 1, scenario 57g). Sensitivity calculations involve the same total volume of β-NAT (i.e. condensed $HNO_3$) and mode radii of 3.0 µm and 1.0 µm,

respectively (Fig. 19).

The results show that for AR=0.1 the spectral signature around 820 cm$^{-1}$ becomes increasingly "peak-like" for mode radii decreasing from 4.8 µm to 1.0 µm (Fig. 20a, 20c, and 20e, blue). While for r=4.8 µm the signature shows a characteristic "shoulder-like" pattern, a superposition of a "shoulder-like" and a "peak-like" signature results for r=3.0 µm. A developed "peak-like" signature as discussed by Höpfner et al. (2006a) is found for r=1.0 µm, and the simulated spectra are almost

identical to the corresponding Mie scenario for both channels (Fig. 20e and 20f) except for slightly higher radiances below ~860 cm-1 in for the AR=0.1 scenario. Finally, the Mie calculations show that a modified "shoulder-like" signature around 820 cm$^{-1}$ along with further differences from the AR=0.1 scenario can be modelled for spherical particles with r=3.0 µm.

### Acknowledgements

The authors thank Myasishchev Design Bureau and the ESSenCe coordination team for a successful *Geophysica* field

campaign. ESSenCe was supported by the European Space Agency/Mission Science Division under the ESSenCe project (Technical Assistance for the Deployment of Airborne Limbsounders during ESSenCe). W. Woiwode is grateful to the Karlsruhe House for Young Scientists for supporting a 5-month research stay at NASA Langley airborne Research Center (NASA LaRC, Hampton, USA), and thanks M. Pitts and L. Poole from the CALIPSO PSC team for a great and productive time at NASA LaRC. We thank M. I. Mishchenko (NASA Goddard Institute for Space Studies, New York, USA) for helpful

recommendations and providing the contact to L. Bi (Department of Atmospheric Sciences, Texas A&M University, College Station, USA, now at: School of Earth Sciences, Zhejiang University, Hangzhou, China), who performed the extensive IIM+SOV T-Matrix calculations. For calculations for moderately aspheric particles, we used the double-precision T-Matrix code for randomly oriented nonspherical particles provided by M. I. Mishchenko, L. D. Travis, and D. W. Mackowski at http://www.giss.nasa.gov/staff/mmishchenko/t_matrix.html. The dataset for the simulation of highly aspherical β-NAT

particles used here is available by L. Bi (bilei@zju.edu.cn). We thank U. M. Biermann, L. J. Richwine, R. F. Niedziela and A.

Y. Zasetsky for providing the refractive indices of β-NAT, α-NAT, STS, NAD and ice. The CLaMS simulation was performed using computing time granted on the supercomputer JUROPA at Jülich Supercomputing Centre (JSC) under the VSR project ID JICG11. We thank EMCWF for the data used for the MIPAS-STR retrievals, the CLaMS simulation and the potential vorticity map. We thank Wyoming Atmospheric Soundings (Department of Atmospheric Science, University of Wyoming, USA) for providing the radiosonde data (see http://weather.uwyo.edu/upperair/sounding.html). We acknowledge the Physical Sciences Division, Earth System Research Laboratory, NOAA, Boulder, Colorado, USA, for providing the sea surface temperature data (see http://www.esrl.noaa.gov/psd/). We thank R. Müller (Institute of Energy and Climate Research (IEK-7), Forschungszentrum Jülich GmbH, Germany) for helpful comments. We thank two anonymous referees, H. Grothe and M. J. Rossi for helpful comments. We acknowledge support by the Deutsche Forschungsgemeinschaft and Open Access Publishing Fund of the Karlsruhe Institute of Technology.

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

Table 1. Overview of radiative transfer scenarios.

| ID | Species | Size distribution | Code / aspect ratio (AR) | Vertical PSC extent [km] | Vertical range of trop. cloud [km] / $\alpha$ [km$^{-1}$] | Comment |
|---|---|---|---|---|---|---|
| 43 | β-NAT | B | Mie[a] / 1.0 | 17-23 | 0-10 / 0.0223 | |
| 43a | β-NAT | B | Mie[a] / 1.0 | 17-23 | 0-10 / 0.0223 | no scattering |
| 44 | β-NAT | A | Mie[a] / 1.0 | 17-23 | 0-10 / 0.0223 | |
| 45 | STS | B | Mie[a] / 1.0 | 17-23 | 0-10 / 0.0223 | |
| 46 | STS | A | Mie[a] / 1.0 | 17-23 | 0-10 / 0.0223 | |
| 47 | NAD | B | Mie[a] / 1.0 | 17-23 | 0-10 / 0.0223 | |
| 48 | NAD | A | Mie[a] / 1.0 | 17-23 | 0-10 / 0.0223 | |
| 49 | α-NAT | B | Mie[a] / 1.0 | 17-23 | 0-10 / 0.0223 | |
| 50 | α-NAT | A | Mie[a] / 1.0 | 17-23 | 0-10 / 0.0223 | |
| 51 | Ice | B | Mie[a] / 1.0 | 17-23 | 0-10 / 0.0223 | |
| 52 | Ice | A | Mie[a] / 1.0 | 17-23 | 0-10 / 0.0223 | |
| 53 | β-NAT | B | TM[b] / 0.5 | 17-23 | 0-10 / 0.0223 | |
| 54 | β-NAT | A | TM[b] / 0.5 | 17-23 | 0-10 / 0.0223 | |
| 55 | β-NAT | B | TM[b] / 2.0 | 17-23 | 0-10 / 0.0223 | |
| 56 | β-NAT | A | TM[b] / 2.0 | 17-23 | 0-10 / 0.0223 | |
| 57 | β-NAT | B | TM[c] / 0.1 | 17-23 | 0-10 / 0.0223 | |
| 57a | β-NAT | B | TM[c] / 0.1 | 17-23 | 0-10 / 0.0223 | no scattering |
| 57b | β-NAT | B | TM[c] / 0.1 | 17-23 | 0-2 / 5.000 | |
| 57c | β-NAT | B | TM[c] / 0.1 | 17-23 | . /. | |
| 57d | β-NAT | B | TM[c] / 0.1 | 17-23 | 0-10 / 5.000 | |
| 57e | β-NAT | B | TM[c] / 0.1 | 17-23 | 0-10 / 0.0223 | $T_{Surface} = 273.2$ K |
| 57f | β-NAT | B | TM[c] / 0.1 | 17-20 | 0-10 / 0.0223 | |
| 57g | β-NAT | B1 | TM[c] / 0.1 | 17-23 | 0-10 / 0.0223 | |
| 57i | β-NAT | B2 | TM[c] / 0.1 | 17-23 | 0-10 / 0.0223 | |
| 58 | β-NAT | A | TM[c] / 0.1 | 17-23 | 0-10 / 0.0223 | |
| 59 | β-NAT | B | TM[c] / 10.0 | 17-23 | 0-10 / 0.0223 | |
| 60 | β-NAT | A | TM[c] / 10.0 | 17-23 | 0-10 / 0.0223 | |

[a]Mie code, Höpfner et al. (2006a)

[b]EBCM T-Matrix code, Mishchenko and Travis (1998)

[c]IIM+SOV T-Matrix code, Bi et al. (2013)

Table 2. Particle size distributions used in radiative transfer scenarios.

| Size distribution | Mode 1 | | | Mode 2 | | |
|---|---|---|---|---|---|---|
| | $r_1$ [μm] | $\sigma_1$ | $N_1$ [cm$^{-3}$] | $r_2$ [μm] | $\sigma_2$ | $N_2$ [cm$^{-3}$] |
| A | 2.5 | 1.5 | $1.6 \cdot 10^{-3}$ | 6.0 | 1.35 | $1.5 \cdot 10^{-3}$ |
| B | 2.0 | 1.5 | $1.6 \cdot 10^{-3}$ | 4.8 | 1.35 | $1.5 \cdot 10^{-3}$ |
| B1 | ./. | ./. | ./. | 4.8 | 1.35 | $1.5 \cdot 10^{-3}$ |
| B2 | 0.5 | 1.5 | $1.6 \cdot 10^{-1}$ | 4.8 | 1.35 | $1.5 \cdot 10^{-3}$ |

Table 3. Comparison of equivalent $HNO_{3(g)}$ corresponding with the particle size distributions used in the radiative transfer simulations of the MIPAS-STR observations, the size distributions derived from the in-situ measurements, and a CLaMS simulation, together with excess $HNO_{3(g)}$ derived for the nitrification layer below the observed PSC.

| Size Distribution | $HNO_{3(g)}$ [ppbv] | Comment |
|---|---|---|
| A | 18.2 | radiative transfer simulation, starting point |
| B | 9.3 | radiative transfer simulation, optimized scenario |
| B1 | 8.4 | radiative transfer simulation, sensitivity |
| B2 | 9.8 | radiative transfer simulation, sensitivity |
| FSSP-100 | 18.5 | derived from in-situ observation |
| CDP | 28.5 | derived from in-situ observation |
| CLaMS | 1.6 | chemistry transport simulation |
| nitrification layer[a] | 5.0 | excess $HNO_{3(g)}$ @ ~16.0 km |
| | 7.5 | equivalent $HNO_{3(g)}$ @ 18.5 km |

[a]see Sect. 2.5

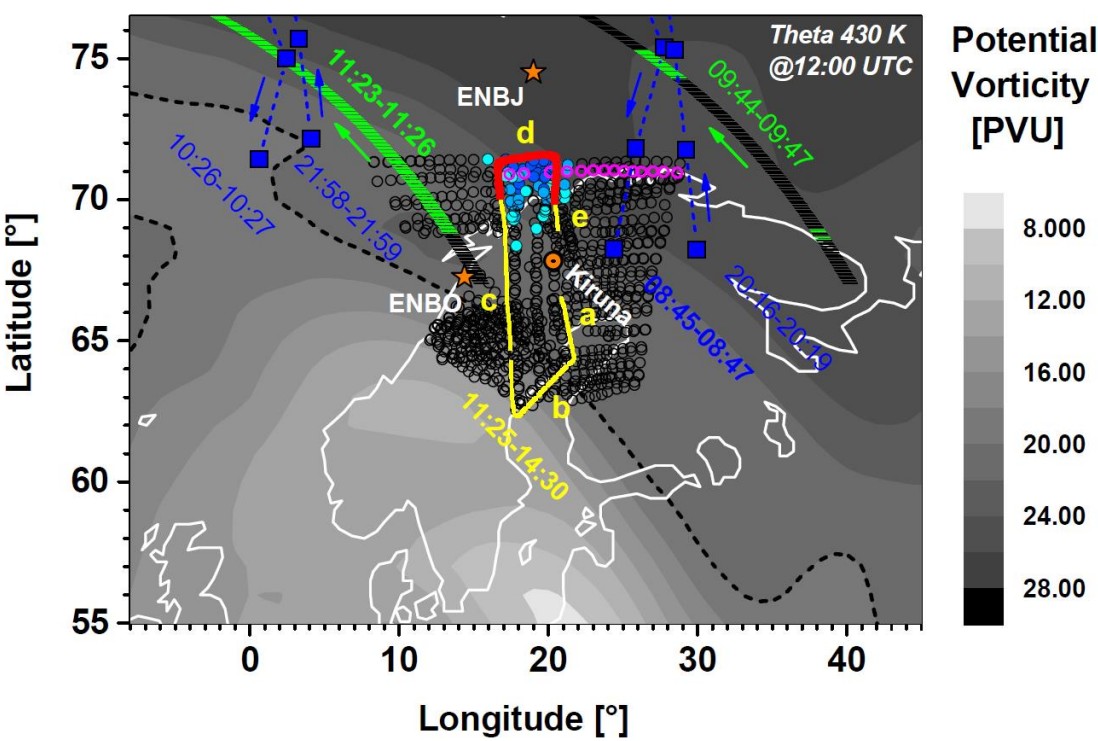

Figure 1. Flight path (yellow line) of the *Geophysica* and meteorological situation during the PSC flight on 11 December 2011. Grey shading: ECMWF ERA-Interim potential vorticity at the potential temperature level of 430 K. Black dashed line: polar vortex edge according to Nash et al. (1996). Open black circles: MIPAS-STR tangent points. Filled circles (cyan to blue): MIPAS-STR tangent points >16 km with cloud index <3. Magenta circles: MIPAS-STR limb scan associated with spectra considered in radiative transfer simulations. Red line: flight section of FSSP-100 and CDP PSC observations. Horizontal green/black bars: CALIPSO PSC observations (green: PSC detected at 21 km). Filled blue squares: MIPAS-Envisat observations at 18.4 to 19.0 km exhibiting PSC signatures. Orange stars: radiosonde launch sites Bodø (ENBO) and Bjornoya (ENBJ).

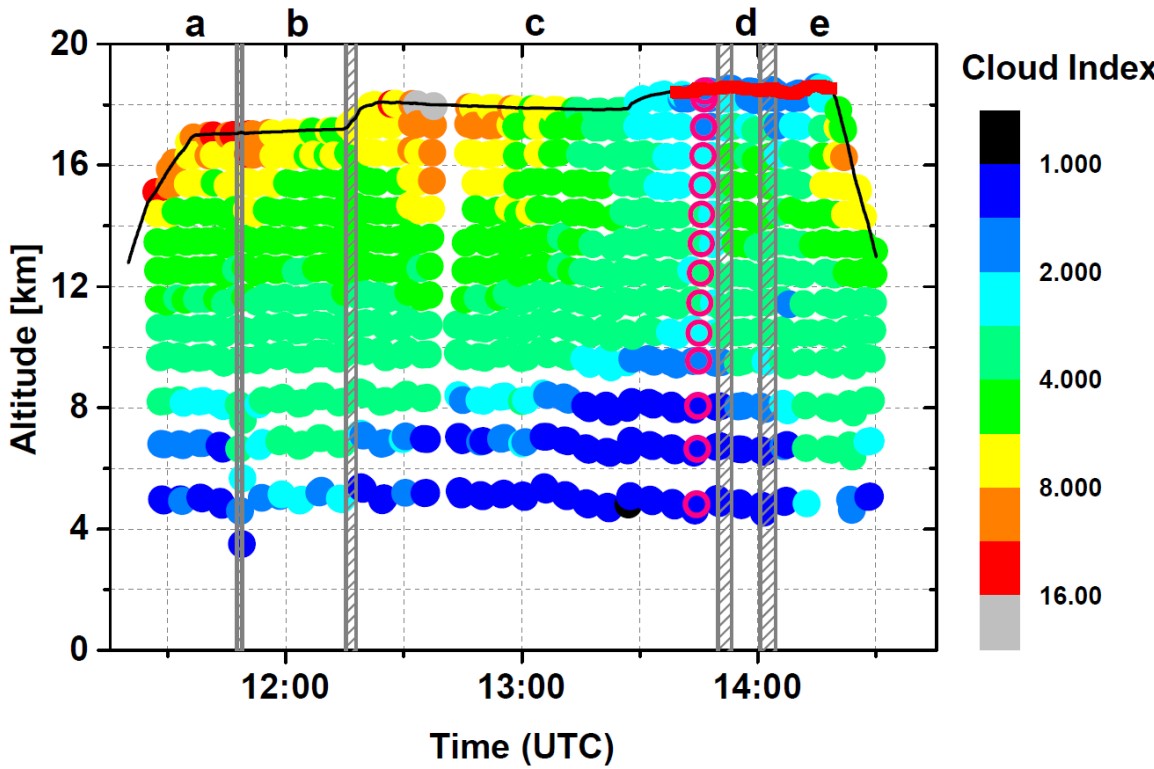

Figure 2. Vertical flight profile of the *Geophysica* (black line) and vertical distribution of the MIPAS-STR tangent points, colour-coded with the cloud index (filled circles). Open magenta circles: MIPAS-STR limb scan associated with spectra considered in radiative transfer simulations. Red line: flight section of FSSP-100 and CDP PSC observations. Grey hatched areas: turns between different legs.

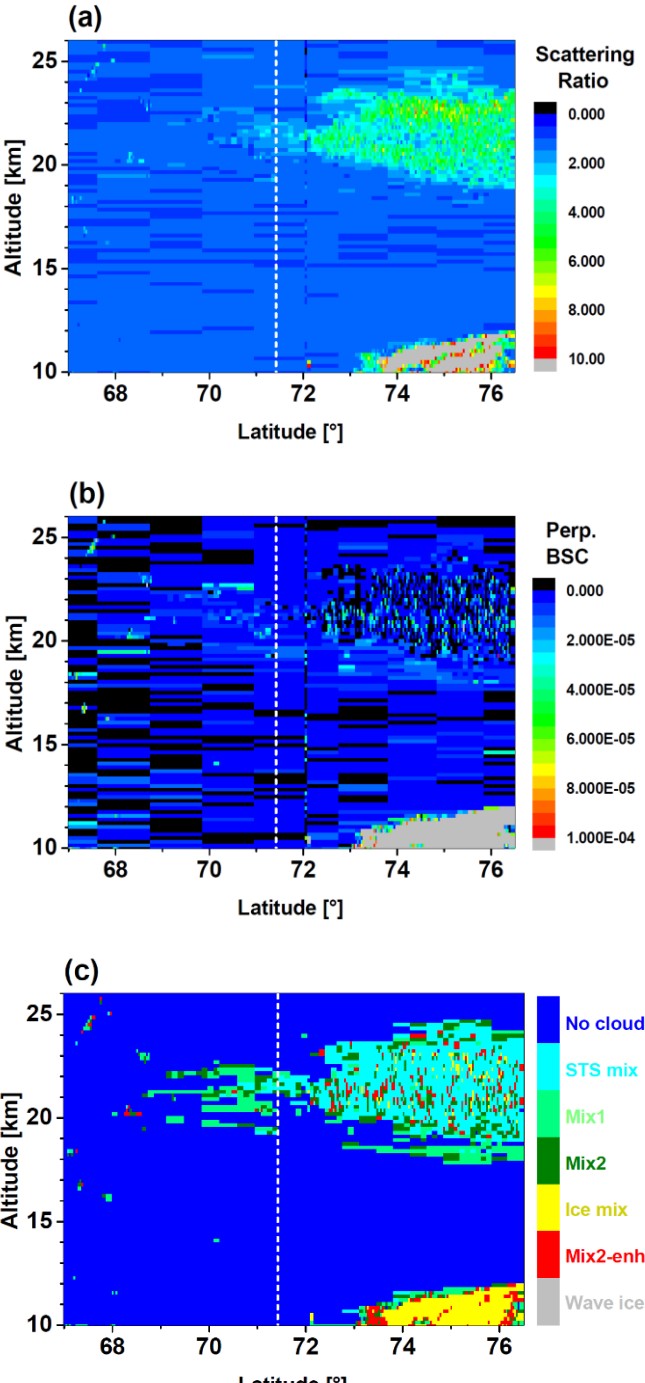

Figure 3. CALIPSO PSC observations associated with the *Geophysica* flight on 11 December 2011. **(a)** Scattering ratio at 532 nm. **(b)** Perpendicular backscatter (BSC) at 532 nm. **(c)** PSC classification according to Pitts et al. (2011, 2013). White dashed lines: latitude of northernmost observations aboard the *Geophysica*.

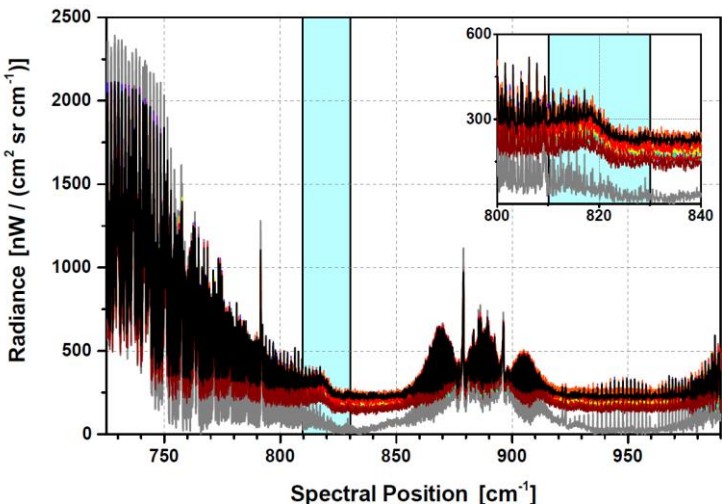

Figure 4a. MIPAS-STR channel 1 spectra of approximately horizontal limb views. Coloured spectra: observations inside PSC in the time interval of the in-situ PSC observations. Black spectrum: PSC observation analysed in Sect. 3 (elevation angle -0.25° versus horizon, flight altitude 18.507 km, tangent altitude 18.445 km). Grey spectrum: Corresponding observation under cloud-free conditions. Spectral region of "shoulder-like" signature around 820 cm$^{-1}$ attributed to β-NAT particles shaded in cyan.

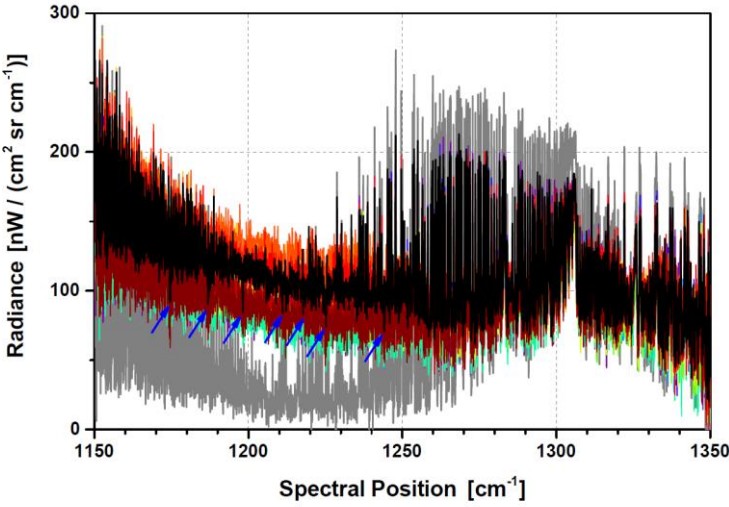

Figure 4b. Same as Figure 4a but for MIPAS-STR channel 2. Blue arrows denote prominent $H_2O_{(g)}$ absorption signatures from scattering of tropospheric radiation by PSC particles.

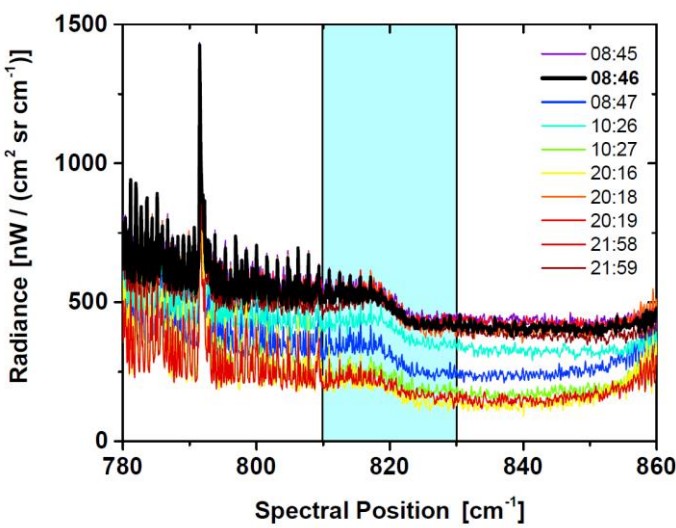

Figure 5. Spectral window of MIPAS-Envisat channel A observations with tangent altitudes between 18.4 and 19.0 km in the vicinity of the *Geophysica* flight on 11 December 2011. Geolocations are indicated in Figure 1. Observation times are given in UTC.

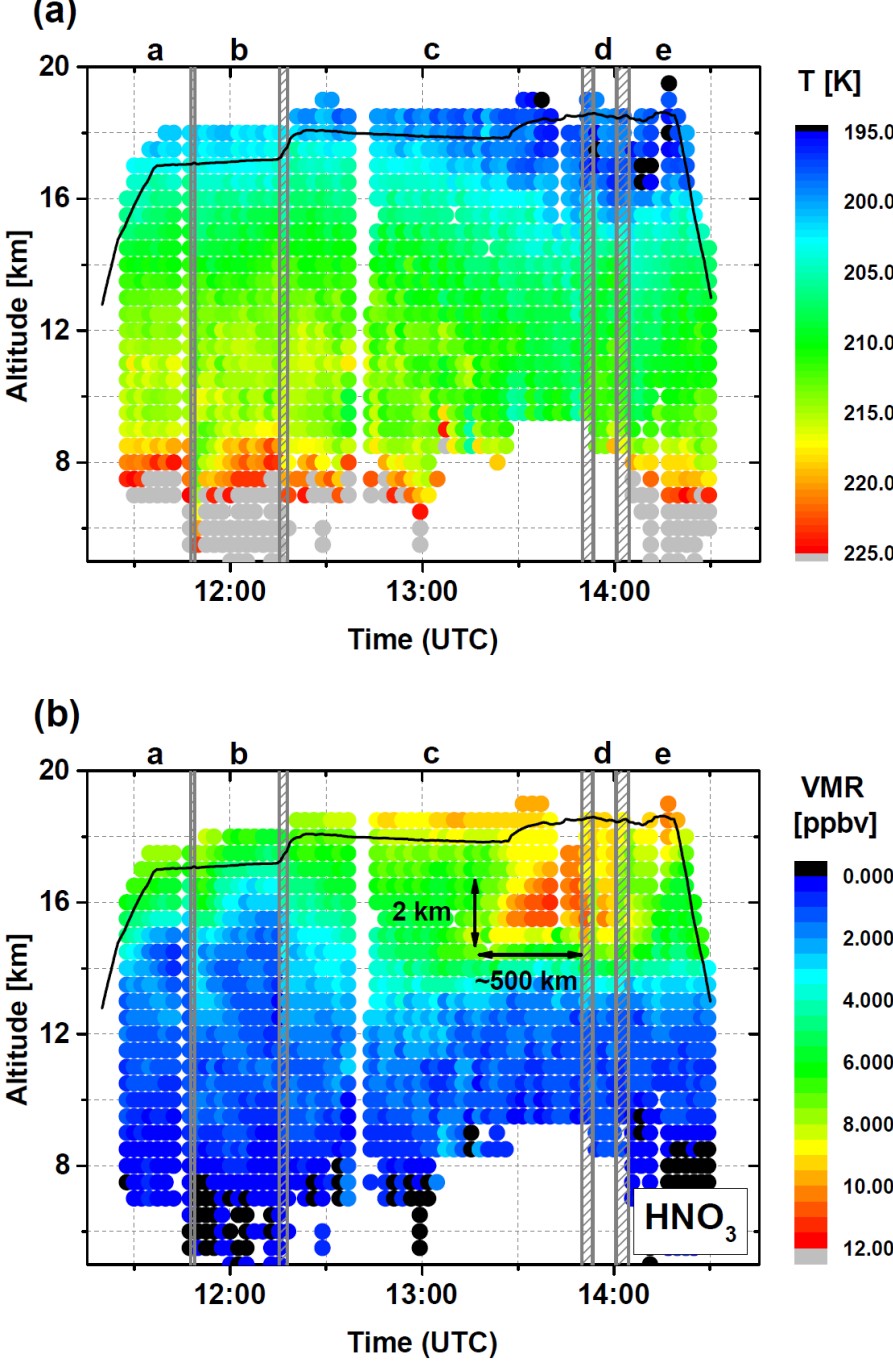

Figure 6. Vertical cross-sections of temperature (T) and $HNO_{3(g)}$ along flight track retrieved from the MIPAS-STR observations. Vertical flight profile (black line) and turns between different flight legs (grey hatched areas). Data points are filtered for a vertical resolution better than 5 km.

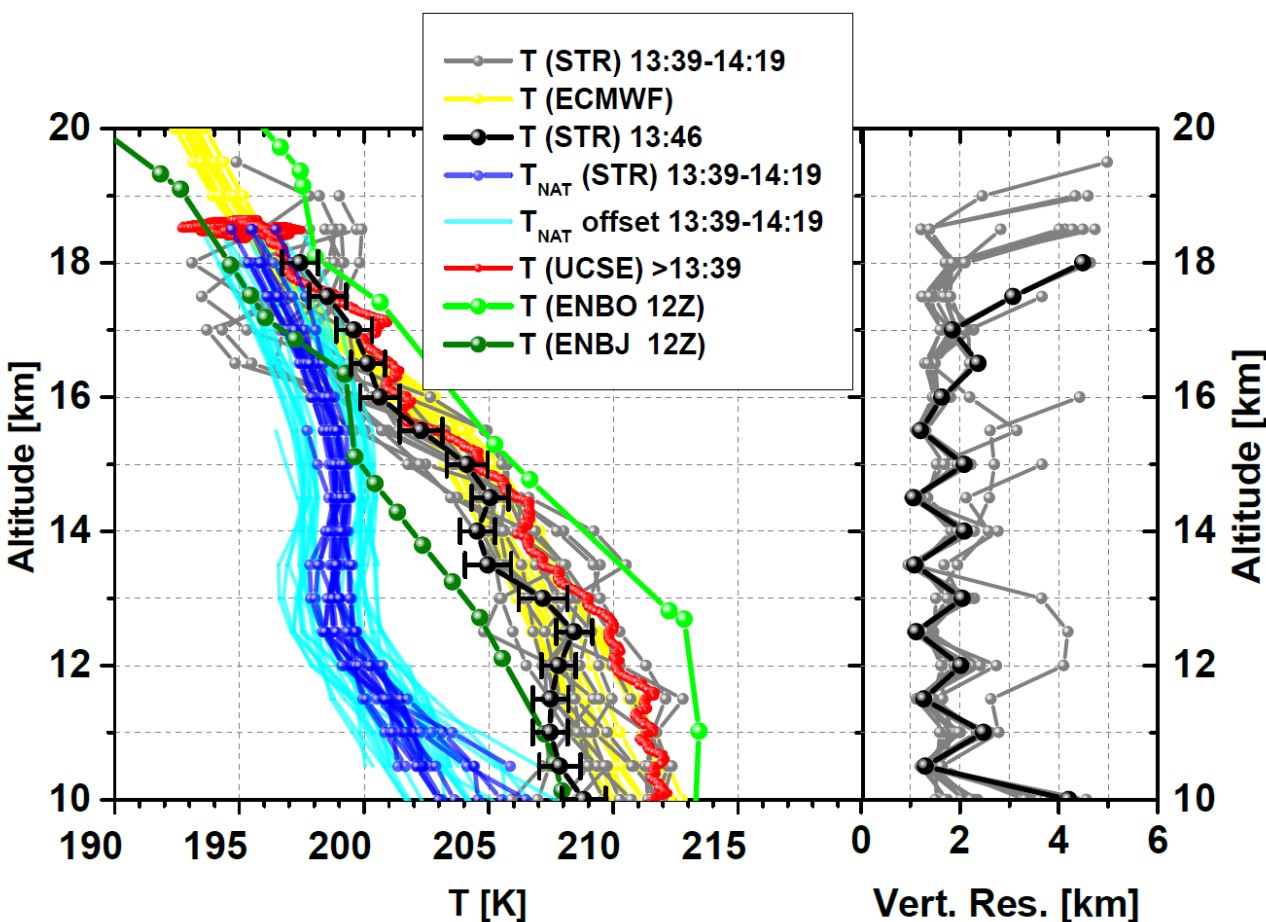

Figure 7. Left panel: MIPAS-STR temperature profiles (grey) associated with the *Geophysica* PSC encounter. Associated ECMWF initial guess/a priori profiles (yellow). Individual MIPAS-STR temperature profile (black) with combined random and systematic 1σ-uncertainties. In-situ measurements by the *Geophysica* UCSE onboard temperature sensor (red) during the PSC encounter and the subsequent descent phase. Radiosonde temperature profiles from Bodø (ENBO, green) and Bjornoya (ENBJ, dark green). $T_{NAT}$ profiles calculated from the MIPAS-STR $HNO_{3(g)}$ and $H_2O_{(g)}$ profiles during the PSC encounter (blue) together with the same profiles (cyan) considering a simultaneous +20% and a simultaneous -20% offset for both species, respectively. Right panel: Vertical resolutions of MIPAS-STR profiles.

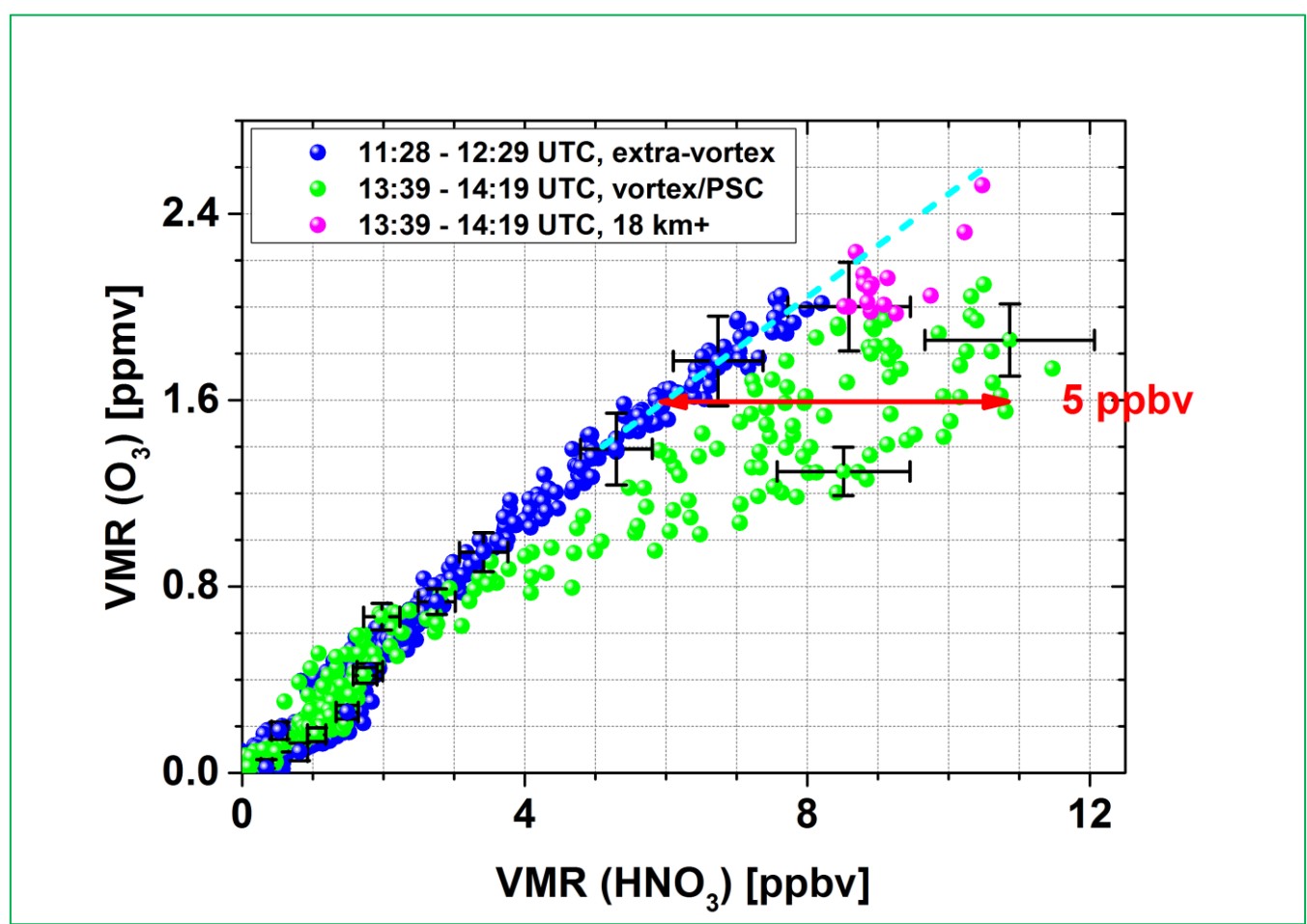

Figure 8. Correlation of $O_3$ with $HNO_{3(g)}$ retrieved from the MIPAS-STR observations together with combined random and systematic $1\sigma$-uncertainties of example data points. Blue data points: correlation outside the polar vortex. Green data points: correlation inside the polar vortex and inside/below the observed PSC (data points only where PSC sufficiently transparent for trace gas retrieval, compare Figs. 6b and 17b). Magenta data points are a subset of the vortex data points close to flight altitude ($\geq 18$ km) inside the PSC. Cyan dashed line: extrapolation of extra-vortex correlation to higher mixing ratios (i.e. altitudes).

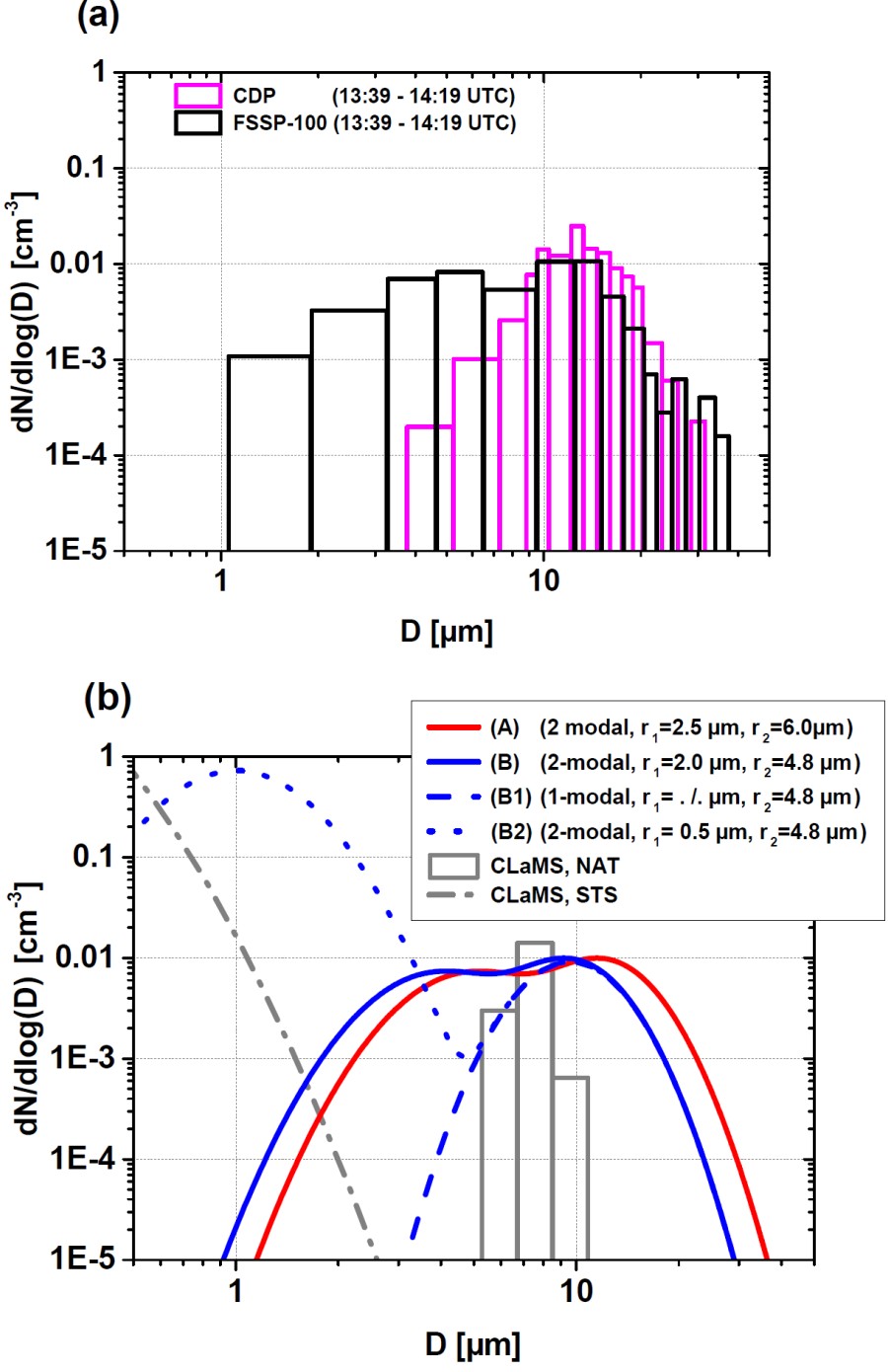

Figure 9. (a) Particle size distributions derived from the FSSP-100 (black) and CDP (magenta) measurements during the PSC encounter (from Molleker et al., 2014, Fig. 12). (b) Particle size distributions used in radiative transfer simulations of MIPAS-STR PSC spectra (red and blue) and simulated by the CLaMS (grey). Note that particle sizes are indicated in diameter (D).

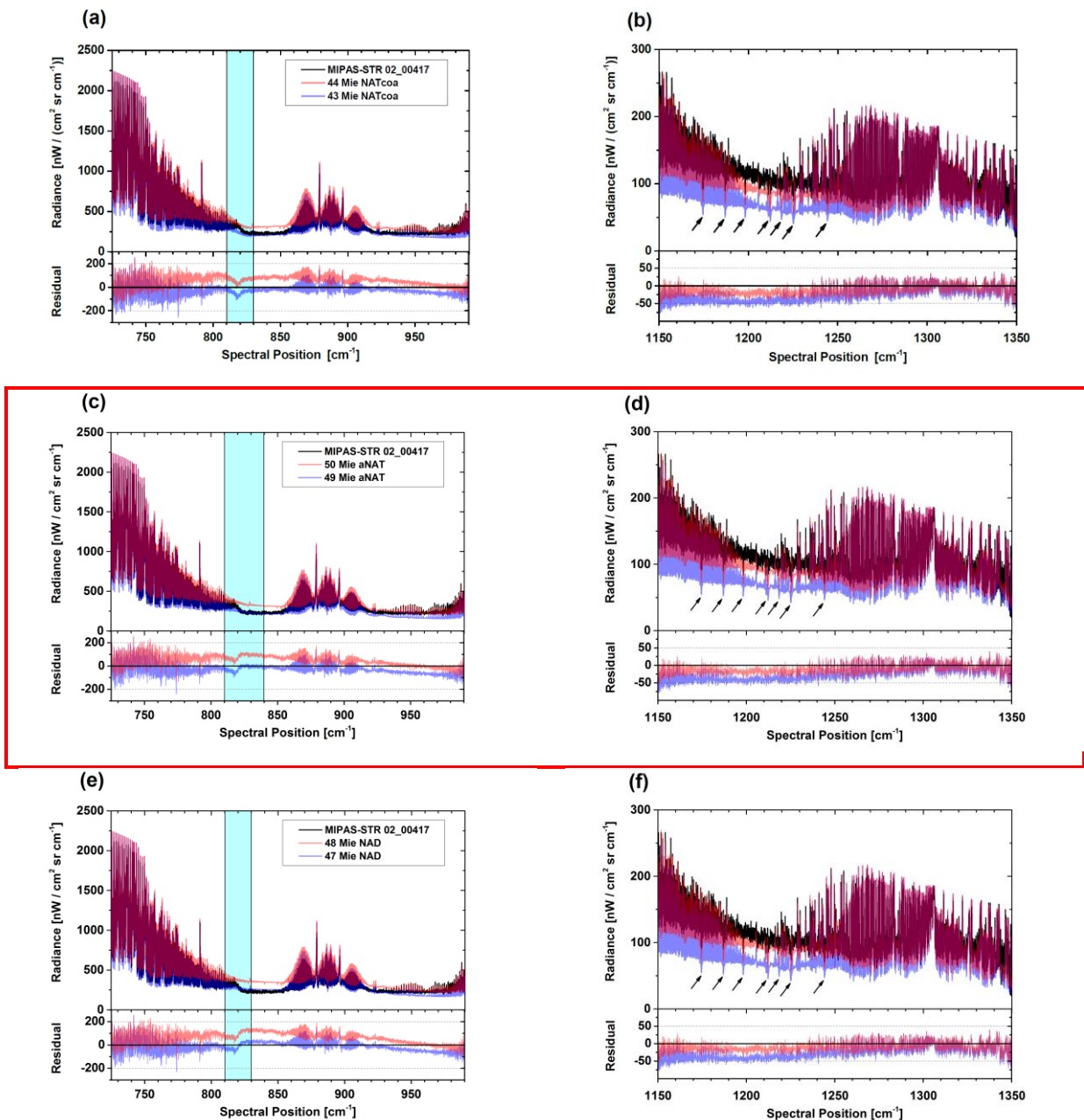

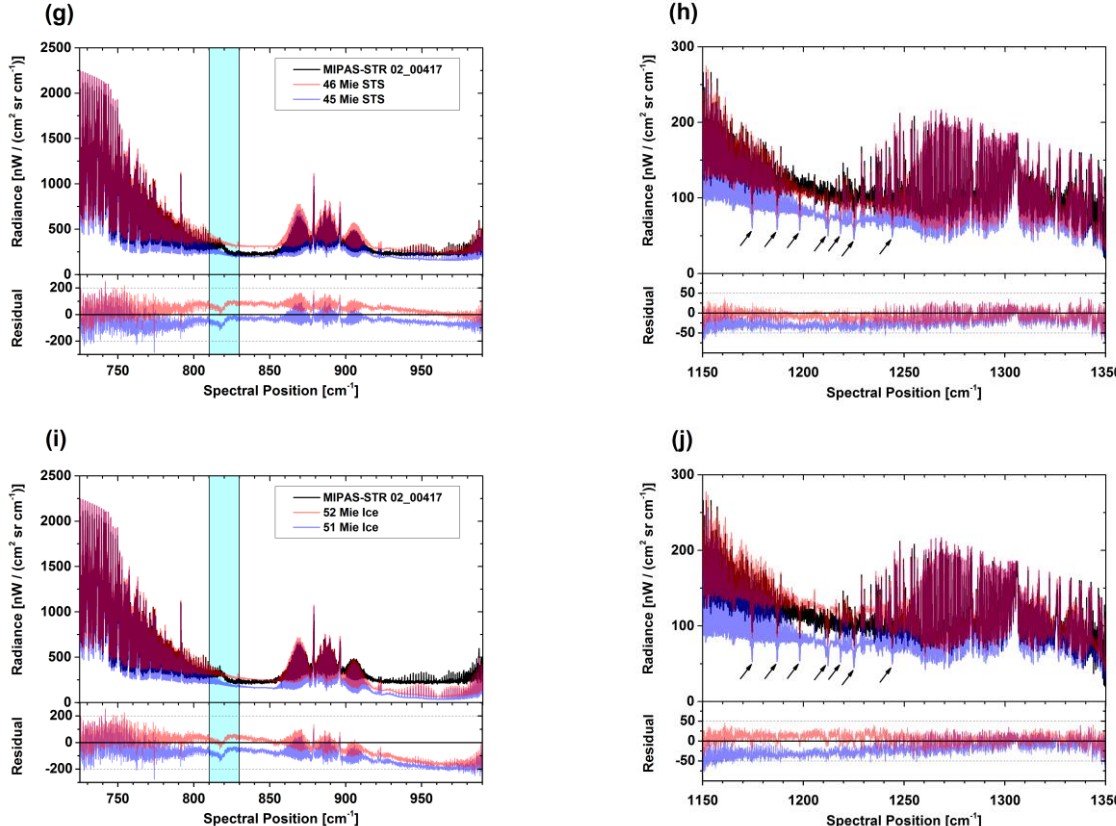

Figure 10. Mie calculations for spherical particles. Black spectra in all panels: MIPAS-STR limb observation (13:47 UTC, tangent altitude 18.4 km). Simulations of  (a) and (b) β-NAT, (c) and (d) α-NAT, (e) and (f) NAD, (g) and (h) STS, and (i) and (j) ice with size distributions A (red) and B (blue). Lower panels: residuals between the simulations and the observation. Black arrows denote prominent $H_2O$ absorption signatures from scattering of tropospheric radiation by PSC particles. Numbers in legend refer to scenarios in Table 1. 'NATcoa' corresponds with β-NAT and 'Mie' with Mie simulation (AR=1.0).

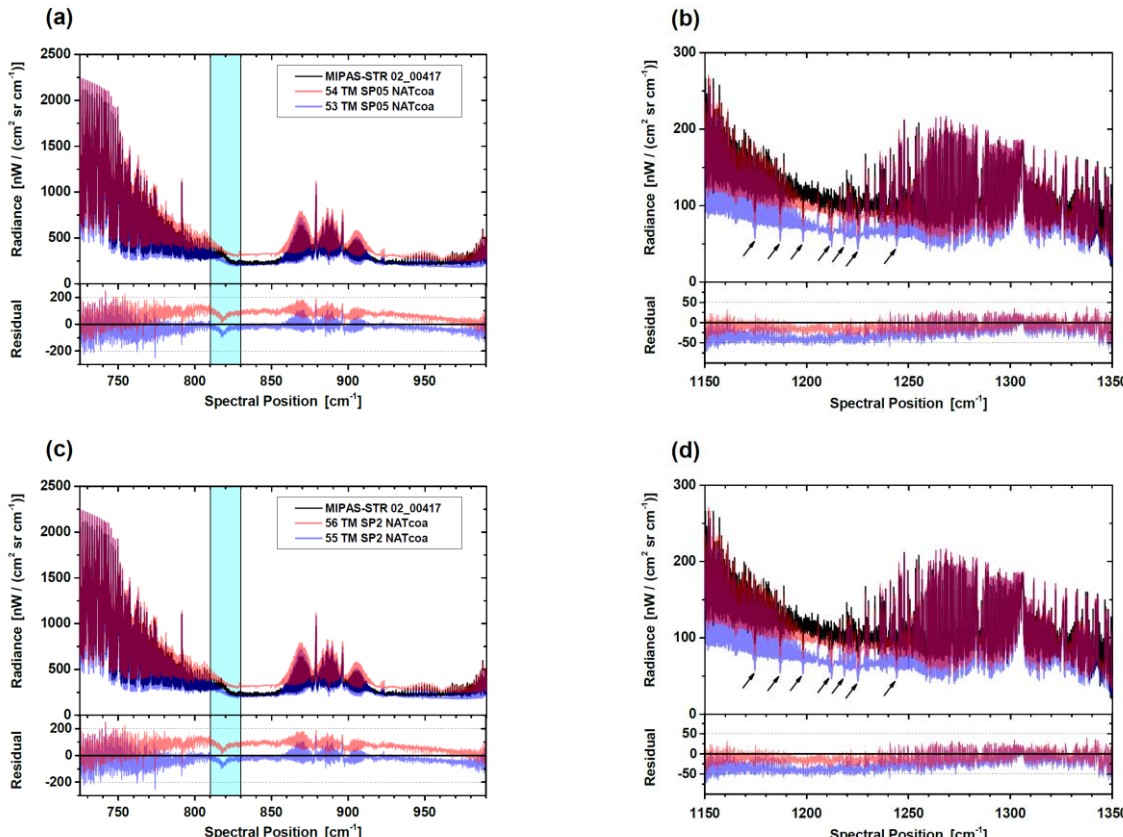

Figure 11. T-Matrix calculations for moderately aspherical β-NAT particles. Black spectra in all panels: MIPAS-STR limb observation (13:47 UTC, tangent altitude 18.4 km). Simulations of β-NAT with (a) and (b) AR=0.5, and (c) and (d) AR=2.0 with size distributions A (red) and B (blue). Lower panels: residuals between the simulations and the observation. Black arrows denote prominent $H_2O$ absorption signatures from scattering of tropospheric radiation by PSC particles. Numbers in legend refer to scenarios in Table 1. 'NATcoa' corresponds with β-NAT, 'TM' with T-matrix simulation and 'SP' with spheroid (numbers indicate AR).

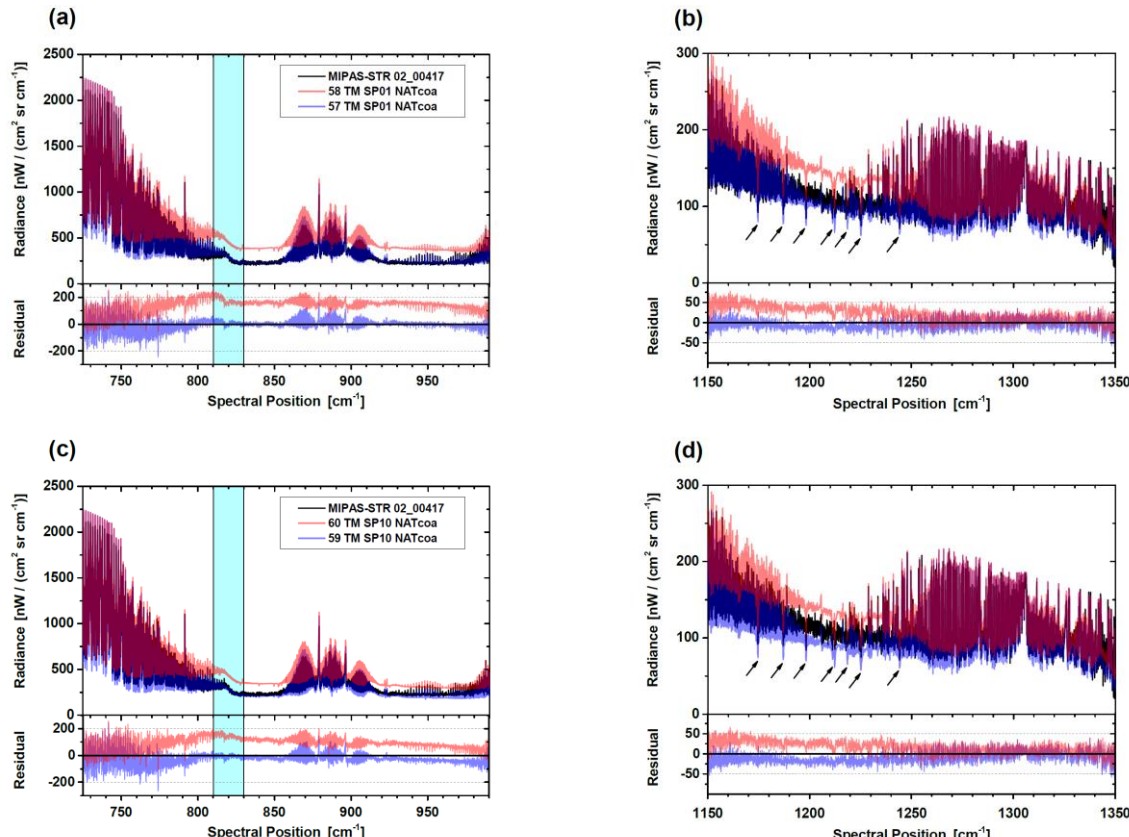

Figure 12. T-Matrix calculations for highly aspherical β-NAT particles. Black spectra in all panels: MIPAS-STR limb observation (13:47 UTC, tangent altitude 18.4 km). Simulations of β-NAT with (a) and (b) AR=0.1, and (c) and (d) AR=10.0 with size distributions A (red) and B (blue). Lower panels: residuals between the simulations and the observation. Black arrows denote prominent $H_2O$ absorption signatures from scattering of tropospheric radiation by PSC particles. Numbers in legend refer to scenarios in Table 1. 'NATcoa' corresponds with β-NAT, 'TM' with T-matrix simulation and 'SP' with spheroid (numbers indicate AR).

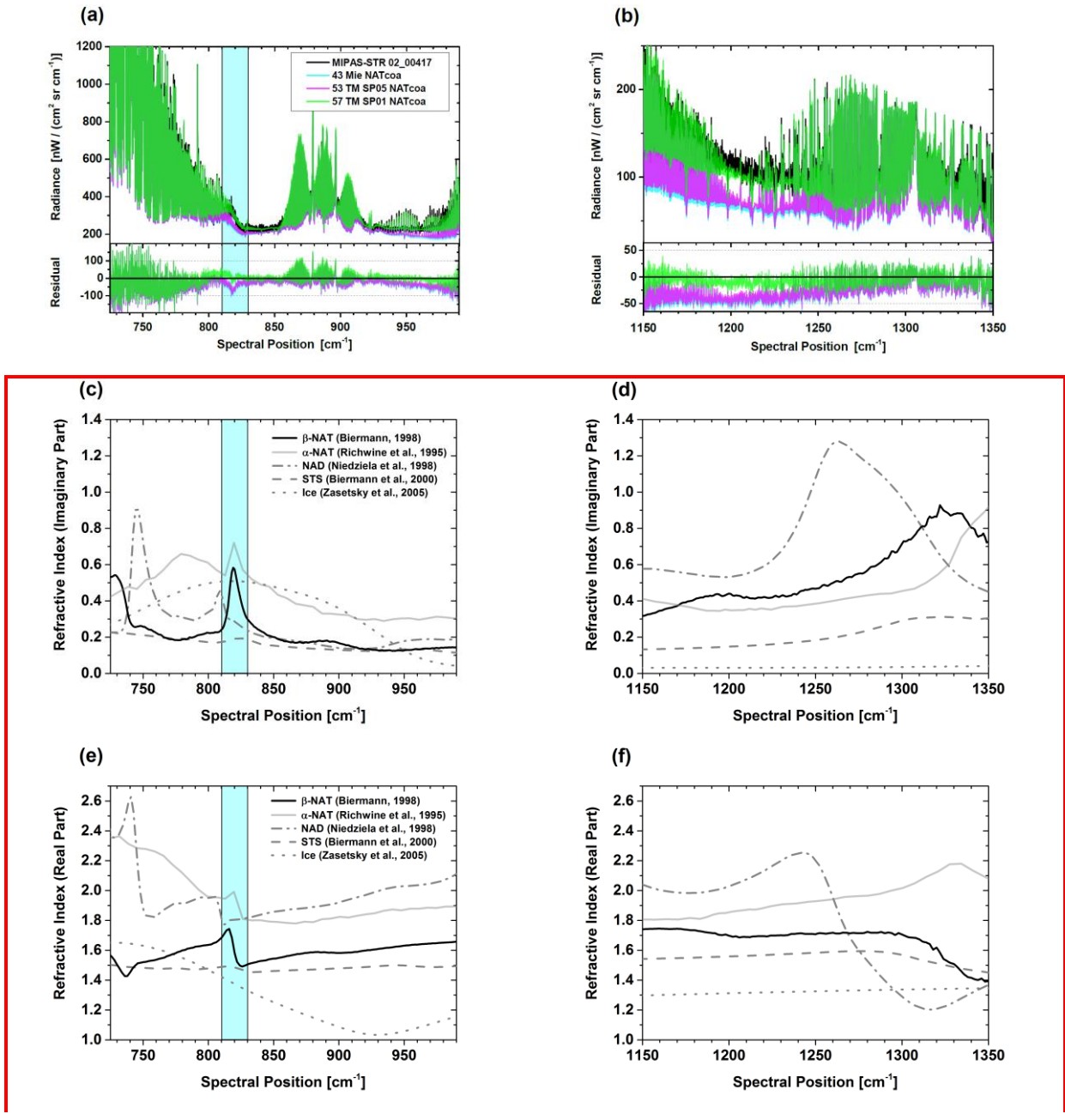

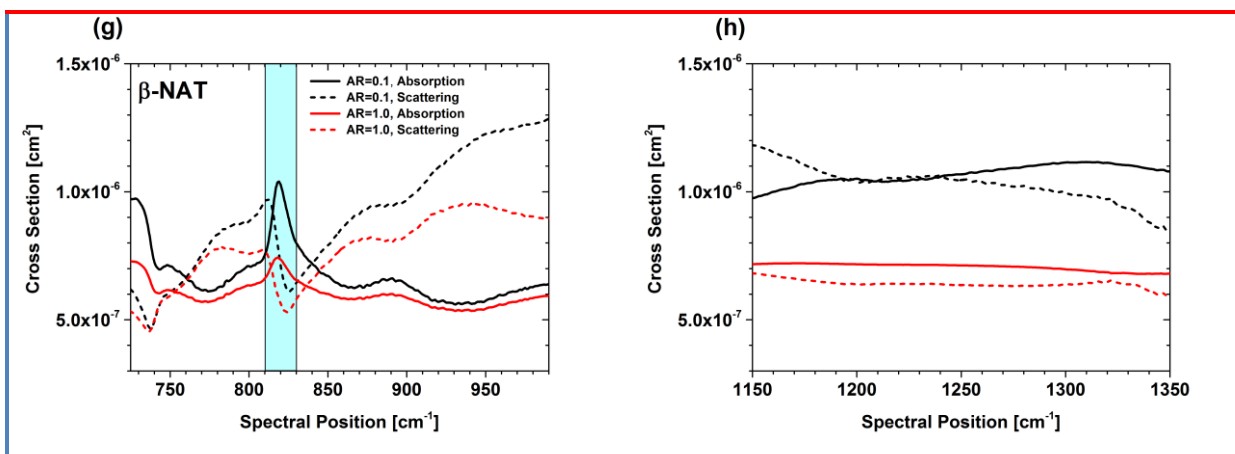

Figure 13. Dependence of simulated spectral signatures of β-NAT particles on degree of non-sphericity. (a) and (b): Black: MIPAS-STR limb observation (13:47 UTC, tangent altitude 18.4 km). Mie calculation for spherical particles (cyan), T-Matrix calculation with AR=0.5 (magenta) and T-Matrix calculation with AR=0.1 (green). 'NATcoa' corresponds with β-NAT, 'Mie' with Mie simulation, 'TM' with T-matrix simulation and 'SP' with spheroid (numbers indicate AR). Lower panels: residuals between the simulations and the observation. Numbers in legend refer to scenarios in Table 1. Refractive index (c) and (d) imaginary and (e) and (f) real parts of β-NAT, α-NAT, NAD, STS and ice. (g) and (h): ensemble-averaged absorption and scattering cross-sections of β-NAT for the discussed AR=0.1 and AR=1.0 scenarios.

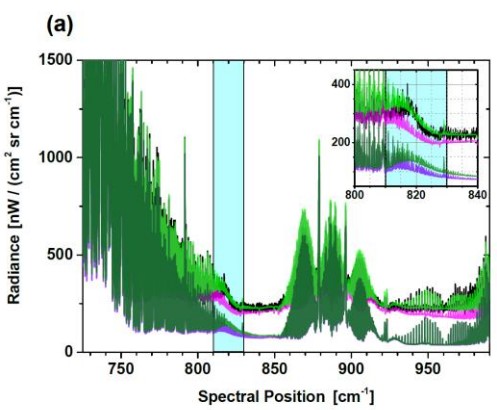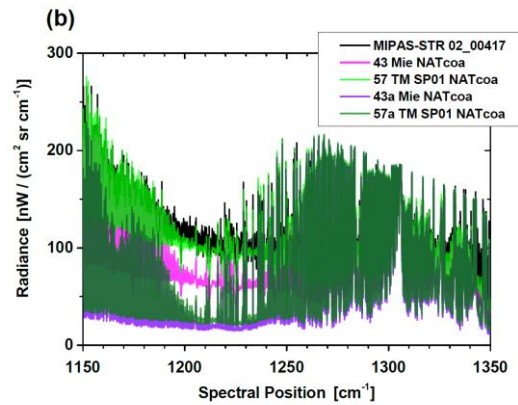

Figure 14. Relative roles of emission and scattering by spherical particles and highly elongated spheroids consisting of β-NAT. Black: MIPAS-STR limb observation (13:47 UTC, tangent altitude 18.4 km). Magenta/violet: simulations of spherical particles (AR=1.0) with/without scattering. Green/dark green: simulations of highly elongated particles (AR=0.1) with/without scattering. Numbers in legend refer to scenarios in Table 1. 'NATcoa' corresponds with β-NAT, 'Mie' with Mie simulation, 'TM' with T-matrix simulation and 'SP' with spheroid (numbers indicate AR).

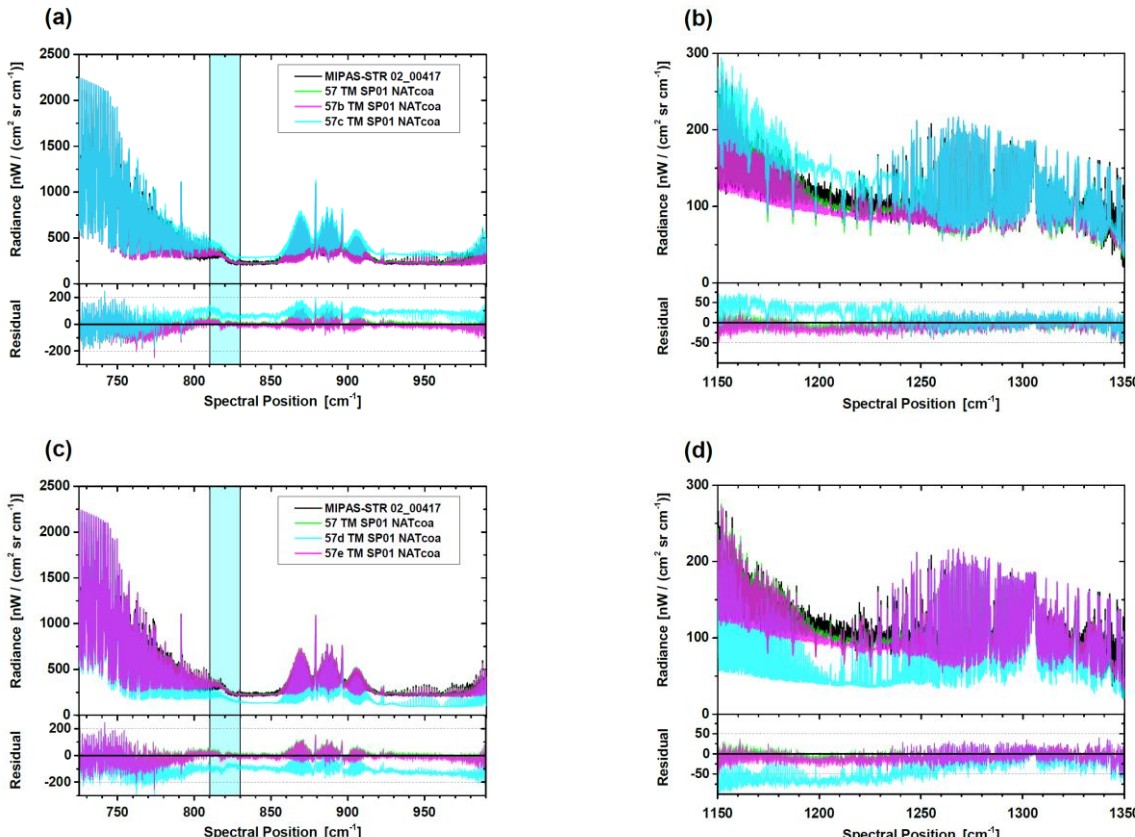

Figure 15. Influence of tropospheric scenario on simulations of elongated spheroids with AR=0.1 consisting of β-NAT. Black spectra in all panels: MIPAS-STR limb observation (13:47 UTC, tangent altitude 18.4 km). Green spectra in all panels (mostly hidden by magenta spectra): optimized scenario. (a) and (b): sensitivity simulations assuming an opaque tropospheric cloud between 0 and 2 km (magenta) and the absence of any tropospheric cloud (cyan). (c) and (d): sensitivity simulations assuming an opaque tropospheric cloud from 0 to 10 km (cyan) and a surface temperature reduced by 7 K (magenta). Numbers in legend refer to scenarios in Table 1. 'NATcoa' corresponds with β-NAT, 'TM' with T-matrix simulation and 'SP' with spheroid (numbers indicate AR).

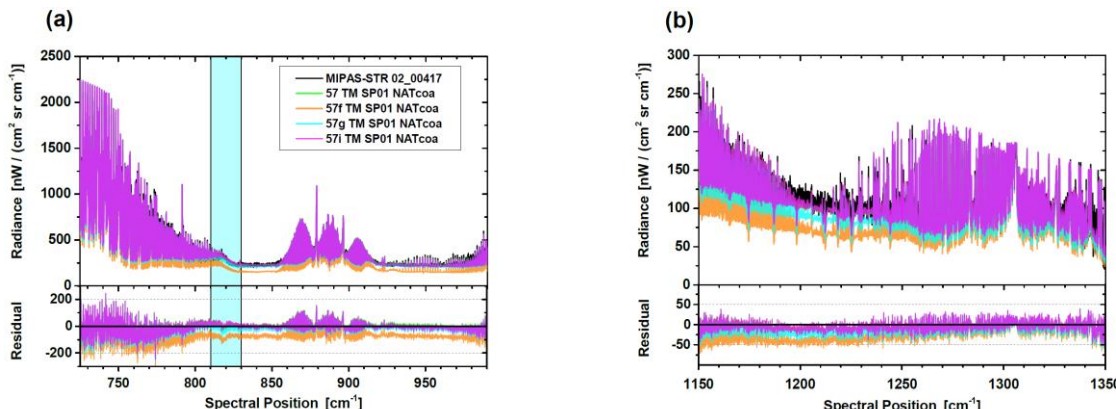

Figure 16. Influence of PSC layer thickness and particle size distribution on simulations of elongated spheroids with AR=0.1 consisting of β-NAT. Black: MIPAS-STR limb observation (13:47 UTC, tangent altitude 18.4 km). Green (mostly hidden by magenta spectra): optimized scenario (assuming a vertical extent of the PSC from 17 to 23 km and size distribution B). Orange: sensitivity simulation assuming a reduced vertical extent of the PSC from 17 to 20 km. Cyan and magenta: sensitivity simulations using size distributions B1 and B2. Numbers in legend refer to scenarios in Table 1. For size distributions see Table 2 and Figure 9b. 'NATcoa' corresponds with β-NAT, 'TM' with T-matrix simulation and 'SP' with spheroid (numbers indicate AR).

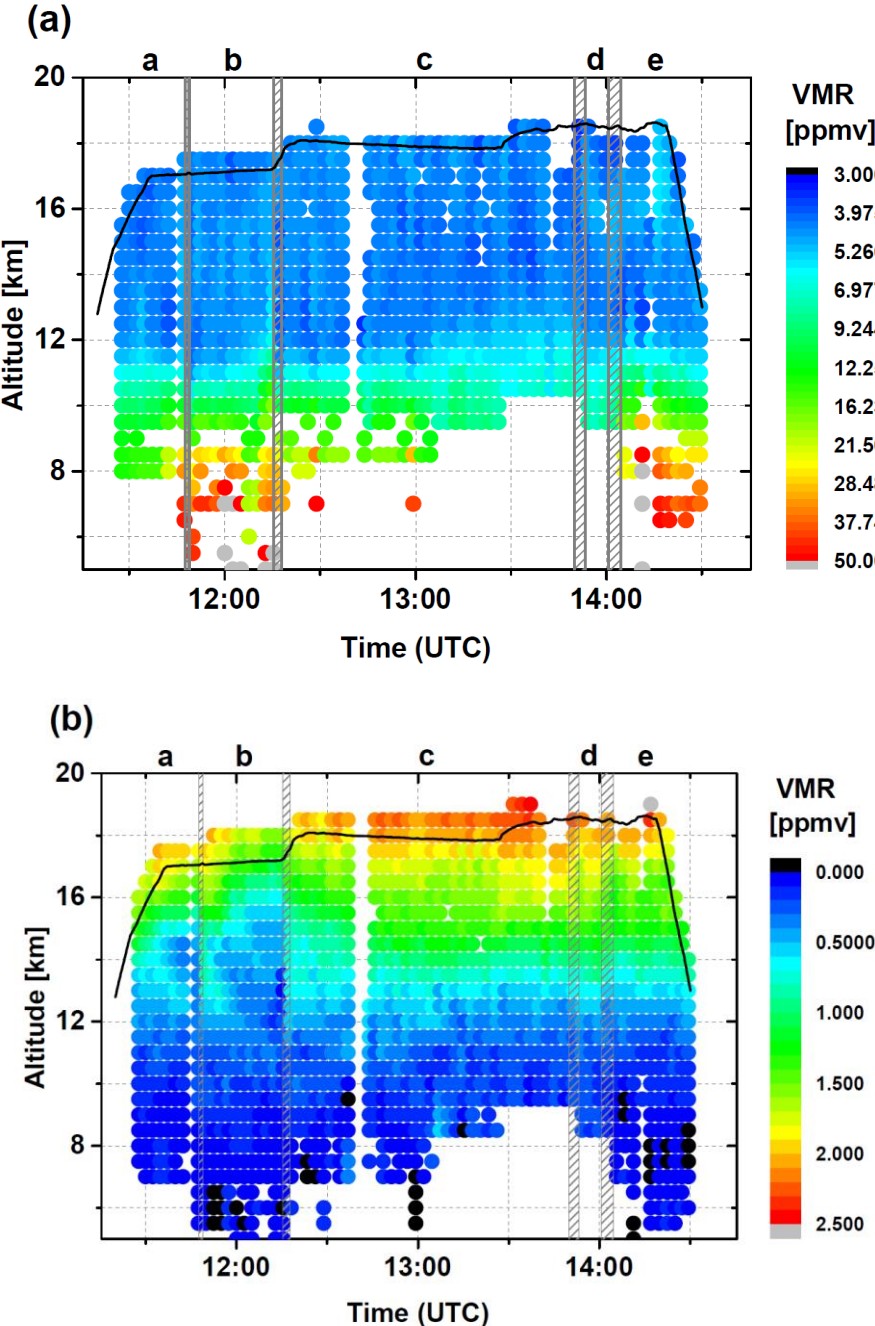

Figure 17. Vertical cross-sections of $H_2O_{(g)}$ and $O_3$ along flight track retrieved from the MIPAS-STR observations. Vertical flight profile (black line) and turns between different flight legs (grey hatched areas). Data points are filtered for a vertical resolution better than 5 km.

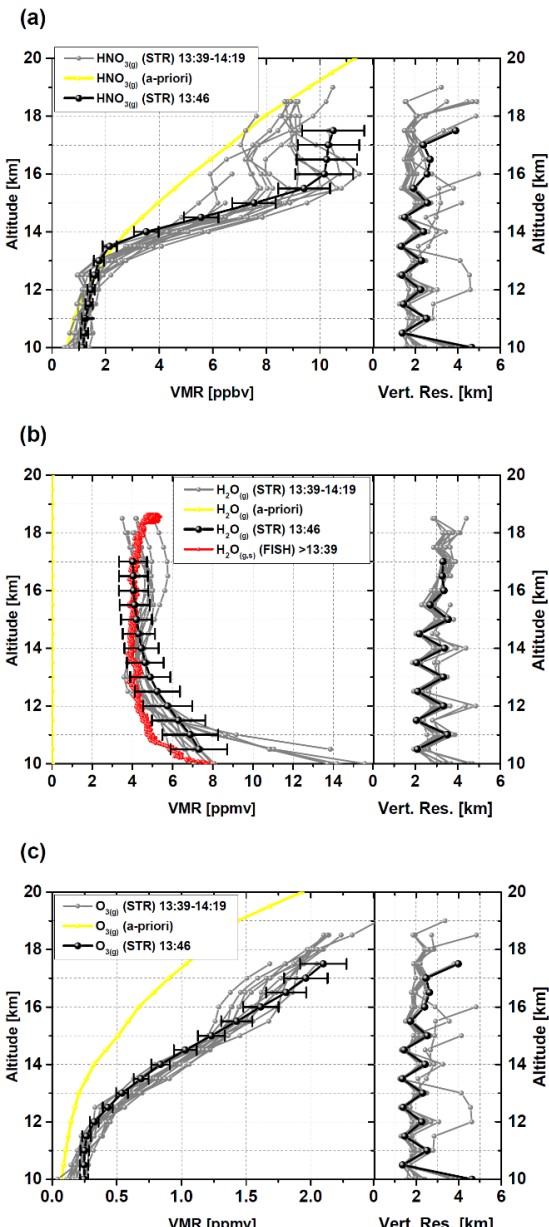

Figure 18. Left panels: Vertical profiles of $HNO_{3(g)}$, $H_2O_{(g)}$ and $O_3$ retrieved from the MIPAS-STR observations during the PSC encounter (grey and black) including initial guess/a priori profiles (yellow; $HNO_{3(g)}$ and $O_3$: Remedios et al., 2007; $H_2O_{(g)}$: 0 ppmv at all altitudes). For $H_2O$, collocated FISH total $H_2O_{(g+s)}$ measurements during the PSC encounter and the subsequent descent phase are shown for comparison. Right panels: Vertical resolutions of MIPAS-STR profiles.

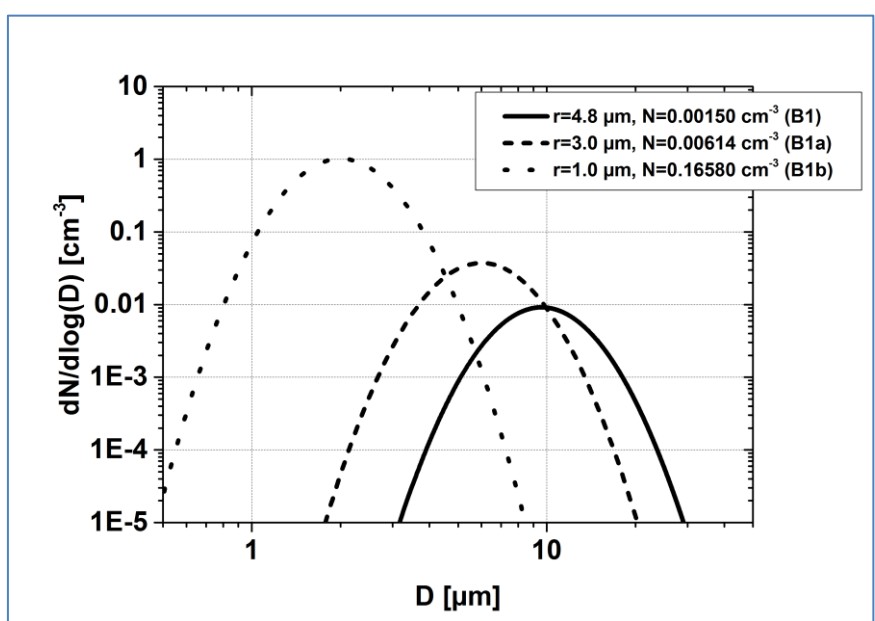

Figure. 19. Size distributions used for sensitivity simulations discussed in Appendix B. The total volume of condensed β-NAT is constant (corresponding with 8.4 ppbv of gas-phase equivalent $HNO_3$) and the mode with is 1.35 in all cases. For the binned size distribution used in the AR=0.1 scenario with r=1.0, the particle number density had to be scaled to 0.17342 $cm^{-3}$ to match 8.4 ppbv of gas-phase equivalent $HNO_3$.

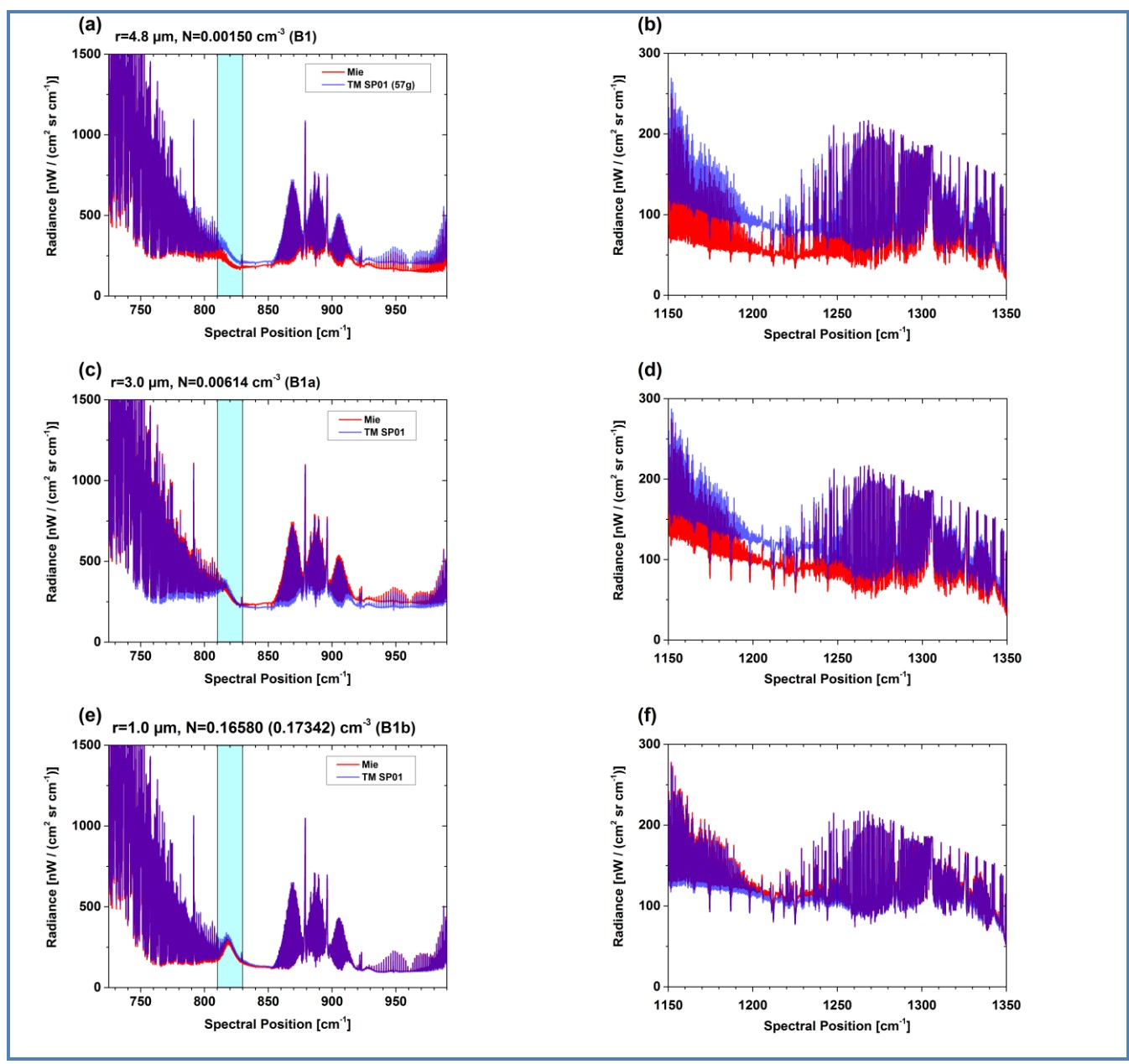

Figure. 20. Sensitivity simulations investigating the effect of decreasing mode radii on the simulated spectral signatures of β-NAT particles for a T-Matrix scenario with AR=0.1 (blue) and a corresponding Mie scenario (AR=1.0, red). (a) and (b): large particle mode, size distribution B1. (c) and (d): intermediate particle mode, size distribution B1a. (e) and (f): small particle mode, size distribution B1b. Numbers above (a), (c) and (e) indicate corresponding mode radii and particle number densities (see Fig. 19). 'TM' corresponds with T-matrix simulation and 'SP' with spheroid (numbers indicate AR).