# Peer review of "Spectroscopic evidence for large aspherical $\beta$ -NAT particles involved in denitrification in the December 2011 Arctic stratosphere"

_Atmospheric Chemistry and Physics, 2016_

## Referee Comment (RC1) · Anonymous Referee #1 · 8 Apr 2016

Woiwode et al. analyze infrared limb emission spectra that were recorded with the airborne MIPAS-STR instrument during a flight above northern Scandinavia in December 2011. The spectra reveal a "shoulder-like" signature at 820 cm-1 which (in a slightly different manner, i.e., more "peak-like") has already previously been observed in space-borne infrared PSC observations and has been assigned to beta-NAT particles. Ambient conditions like temperatures around flight altitude during the PSC encounter support the presence of NAT (temperature is above the existence temperatures of ice and STS). Additionally, a local maximum of gaseous HNO3 was detected just below the PSC encounter which could result from nitrification. From simulations with Mie theory, the authors confirm that only with the refractive indices of beta-NAT it is possible to

at least roughly reproduce the observed signature at 820 cm-1. But only when considering highly aspherical particle shapes with aspect ratios of 0.1 and 10 in T-matrix computations, a satisfactory agreement between MIPAS-STR measurements and simulations is obtained.

General comment

This is a very comprehensive study containing a wealth of information, not only the detailed spectral analysis of the PSC signatures, but also concomitant measurements of particle size distributions (in-situ), trace gas analyses, and space-borne observations. In my opinion, the analysis is sound and indeed gives evidence for the presence of highly aspherical beta-NAT particles due to the much better match of the T-matrix simulations compared to those with Mie theory. The manuscript is well-written, and I have listed only a few points that need further clarification. I therefore support the publication in ACP once the following specific comments are addressed.

Specific comments

1) Page 2, line 27: Here and at some other places in the manuscript (in particular at page 10, line 23ff) no mention is made that there are two IR spectroscopically different modifications of NAD, the low-temperature alpha-NAD and the high-temperature beta-NAD phase (Grothe et al., 2004). The employed optical constants from Niedziela et al. (1998) closely correspond to the alpha-NAD spectrum shown in Fig. 6 in Grothe et al. (2004). In beta-NAD, the v2 (NO3-) band is slightly shifted to higher wavenumbers (811 cm-1, Table 1 in Grothe et al., 2004) and, judging from Fig. 6 (in Grothe et al. 2004), has a higher intensity compared to alpha-NAD. As far as I know, optical constants for beta-NAD have not yet been retrieved, and maybe the small wavenumber shift of the v2 (NO3-) band in beta-NAD would not lead to a better match with the measured MIPAS-STR signature at 820 cm-1 compared to alpha-NAD, but the existence of the high-temperature beta-NAD modification should at least be acknowledged. In annealing experiments, Grothe et al. (2004) observed that the low-temperature alpha-NAD

modification first transformed into beta-NAD at about 200 K, and that decomposition of beta-NAD into beta-NAT and NAM then only occurred at a considerably higher temperature. This finding should also be addressed in the discussion on page 10, line 25.

Grothe, H., Myhre, C. E. L., and Tizek, H., Vibrational spectra of nitric acid dihydrate (NAD), Vib. Spectrosc., 34, 55-62, 2004.

2) Page 3, lines 4/5: Here and/or in the later discussion on page 13, lines 17-19, one could also explicitly mention that Grothe et al. (2006) observed a variety of morphologies for the beta-NAT particles, depending on the growth conditions, supporting the argumentation that a simplified shape assumption might contribute to the discrepancies between the simulations and the observation.

3) Page 8, line 4: See above, the data from Niedziela et al. refer to the signatures of alpha-NAD.

4) Page 8, line 14/15: So size distribution A is a bi-modal log-normal fit to the FSSP-100 observation? One could state this more clearly instead of writing "resembles approximately the size distribution ".

5) Page 12, line 21-24, discussion of Fig. 13: I am wondering whether one could elaborate the effect of particle asphericity on the emission/absorption and scattering of the particles more clearly. From the compilation of refractive indices plotted in Fig. 13 c-f it is clear that the "shoulder-like" signal at 820 cm-1 can only be reproduced by beta-NAT. But also large spherical beta-NAT particles would produce some sort of shoulder-like signal around that wavenumber from the interplay of "peak-like" emission and "step-like" scattering contribution. The important message is that only highly aspherical particles reproduce the correct amplitude of the measured signal. Obviously, the shape dependency is predominantly related to the scattering contribution, because the AR=1.0 and AR=0.1 scenarios without scattering only showed smaller differences. I would propose to include a more fundamental plot that shows the wavenumber-dependent

ensemble-averaged absorption/emission and scattering cross sections for different aspect ratios, so that the reader immediately gets an impression about the change of these basic quantities with particle shape.

6) Page 12, lines 25-27: See above, no distinction between alpha- and beta-NAD.

7) Page 16, line 3-4: Is there any assessment of the influence of particle asphericity on the Mie theory-inferred diameters of the FSSP measurements? There is some discussion of this issue in the conclusion section, but I would propose to directly mention it here where the size distributions from the MIPAS-STR simulations and in-situ observations are compared.

8) Conclusions section: I would like to see a statement/analysis whether there is a certain size threshold above which one can safely infer shape information for the beta-NAT particles from the signature at 820 cm-1. In the introduction, the authors refer e.g. to previous MIPAS-Envisat PSC observations of beta-NAT particles with smaller radii. Here, a "peak-like" rather than a "shoulder-like" signature at 820 cm-1 was observed, probably due to the reduced amount of scattering. When neglecting the scattering source function, however, the shape influence was observed to be less pronounced (Sect. 3.5). So I am wondering whether there are size limitations for the identification of highly aspherical particles from the MIPAS observations.

Technical corrections

1) Page 2, line 11: typo in NO3-

2) Page 5, line 11: CALIOP should be replaced by CALIPSO .

3) Page 9, line 27: I suppose it is meant "by the simulated scattering of radiation", i.e., delete "to".

4) Page 10, line 19: aspherical particles

5) Page 11, lines 18, 23, 24: Please check the units for the given radiances. Shouldn't

it be W cm-2 sr-1 cm?

6) Page 14, line 23: "the first mode is of minor importance"

7) Page 16, line30: "of condensed HNO3."
* * *

---

## Referee Comment (RC2) · Anonymous Referee #2 · 30 Apr 2016

The authors provide a nice analysis of Fourier transform IR spectroscopic measurements in an Arctic PSC in December 2011. The measurements are analyzed under a variety of radiative transfer situations and with a variety of possible PSC size distributions. The authors convincingly demonstrate the signatures of $\beta$-NAT in the data and provide good justification for including the presence of a lower cloud layer and a size distribution with a median radius near 5 $\mu$m in the second mode. The paper is well written and should be published. I have three somewhat major comments.

1) For those not up on the latest PSC literature. How does $\beta$-NAT differ from other NAT? What are the other NAT forms? I assume there is an $\alpha$-NAT. Is there a $\gamma$-NAT?

2) What is the temperature history of the $\beta$-NAT particles sampled? Since the mea-

surement is in December do the presence of these particles provide any information on the possible formation of NAT through heterogeneous nucleation? Did the temperature fall below Tice in the previous days so the nucleation pathways would come through ice? I realize the authors may not wish to pursue this in an already long paper, but it is a natural question which people will be curious about and it may not take much work.

3) The discussion of the differences between the condensed hno3 mixing ratio from the various possible size distributions from the different instruments is confusing. Specifically: In section 4, is there an explanation for why the in situ particle size measurements overestimate the condensed phase nitric acid by about a factor of 2 compared to the MIPAS measurements and compared to what may be realistically expected at these altitudes? Even the MIPAS measurements of ∼9 ppbv are not consistent with the HNO3 gas phase deficit shown in Fig. 18, which appears to be about 2-3 ppbv.

15.8-9 "The bimodal size distribution A corresponds with 18.2 ppbv gas-phase equivalent HNO3 . . . from the FSSP-100 observations, which corresponds to 18.5 ppbv of gas-phase equivalent HNO3". By gas phase equivalent, I assume the authors mean the gas phase mixing ratio should all the condensed phase hno3 be converted to gas phase? If this is the case please state it more explicitly.

This discussion of condensed hno3 is the most confusing of the paper. What is excess hno3, and condensed excess hno3? Excess to what? I assume excess to some predetermined gas phase mixing ratio determined without particles involved. It is not clear how the gray and black hno3 profiles differ in Fig. 18. From the profiles I guess that the black profile is without particles, but this is not stated.

The authors conclude that the hno3 in the MIPAS determined size distributions, ∼8-9 ppbv, is consistent with the gas phase hno3, but would this not mean that all the hno3 is condensed? If I understand Fig. 18 correctly there should be about 2 ppbv hno3 in the particle phase, implying that this is most consistent with CLAMS. The authors should rewrite this section to clarify this discussion.

17.5-10. Here is a plausible explanation for the discrepancy between the in situ and MIPAS estimates of condensed hno3, but this possibility should be described in section 4 where this discrepancy is discussed.

Minor comments:

Figure 2. Why are there no MIPAS circles above the flight altitude of the Geophysica? 7.16, "The close agreement of the temperatures measured by MIPAS-STR, the UCSE data and the ECMWF data with calculated TNAT around flight altitude suggests that the observed PSC was composed of $\beta$-NAT." Does this correspondence really specifically indicate $\beta$-NAT, or just NAT? I have never seen such temperatures specifically target a particular type of NAT. Also what is $\beta$-NAT and how does it differ from ??? What is/are the other option(s)? This should be discussed in the introduction.

8.11 "In Figure 9a, the particle size distributions derived by Molleker et al. (2014) from the FSSP-100 and CDP observations (black and magenta, respectively) during the PSC encounter are shown." This figure shows observations I think, not a derived size distribution. Please clarify.

9.27 "into the simulated spectra by to the simulated" Fix.

10.10-32 and Fig. 10. Do we really need to put the reader through all these Mie calculations since the temperatures are clearly too warm for STS and ice? It seems a waste of space to include Figs 10 e)-h) and the discussion of ice and STS, for all the reasons listed at the bottom of page 10. Just state these reasons up front as to why STS and ice are not considered.

Comment related to most of the Figs 10 -16. A little more effort on the legends in the figure captions would be appreciated. I recognize the space limitations, but the readers do not know what NATcoa means. Why the coa? Nor do the readers know what SP means. I can guess SP10 may mean an aspect ratio of 10 and SP01 an aspect ratio of 0.1, but no directions are provided. Why SP? Use the space to indicate the type of

calculation, the type of particle, the aspect ratio, and if possible the median diameter of the second mode.

13.13-19. Does the fact that the particles may not be homogeneous, throughout the layer sampled, play a role in diluting/spreading the signal? Are there some minimum number of NAT particles required for the signature?

---

## Short Comment (SC1) · 13 May 2016

The investigation by Woiwode et al. (2016) represents an improved and updated reinterpretation of spectroscopic data on the existence of NAT particles in the Artic lower stratosphere. However, the whole evidence is based on the symmetric nitrate deformation band $\nu_2=\delta_s(NO_3^-)$, which is in general the weakest band of the overall mid-infrared spectrum of nitric acid hydrates. Also, the differences between the $\nu_2$ bands of NAT ($\alpha$, $\beta$) and NAD ($\alpha$, $\beta$) are rather small. These differences may also be caused by different crystal structures or different morphologies and textures of the very same phase. This might be worth to be mentioned in the text. Unless concomitant changes in other parts of the mid IR spectrum are not conclusively observed, minor changes in one of the IR absorptions alone constitute weak evidence for the reasons cited above.

The authors have detected highly aspherical NAT particles, which are in accordance with lab experiments (XRD, ESEM) and Avrami calculations by Grothe et al. (2006) and Tizek et al. (2004). The main aspect ratio is 1:10 or 10:1. The authors should however indicate clearly if they assign these ratios to needles or platelets? Are optical calculations able to distinguish between these alternatives?

Only recently Weiss et al. (2016) have shown that $\alpha$-NAT has a much lower heterogeneous nucleation barrier than $\beta$-NAT, where ice acts as the heterogeneous nucleus. In a lab experiment, $HNO_3$ was deposited from the gas phase onto an ice film and the outcome was a pure $\alpha$-NAT film on top of the ice substrate – similar findings are also reported by Gao et al. (2015). This is important in view of the here presented findings, since Grothe et al (2006) found long needles with an aspect ratio of 1:10 only in the presence of ice. On the other hand platelets with a much smaller aspect ratio were found in experiments without ice. So the straightforward conclusion would be that the history of the here observed NAT particles should have involved a heterogeneous nucleation step of $\alpha$-NAT on ice and a subsequent transformation of $\alpha$-NAT into $\beta$-NAT. It would be interesting to note if the presence of ice can be confirmed by limb emission spectra.

Additionally, Tizek et al. (2004) found a stabilization of $\alpha$-NAT in ice matrices. The phase transition was so much hindered by the ice that at about 200 K $\alpha$-NAT could still be observed for several hours. It might be worth to check for the presence of $\alpha$-NAT in the here presented spectra. Iannarelli and Rossi (2016) have estimated an enthalpy difference of 6 ± 20 kJ/mol between $\alpha$- and $\beta$-NAT in favor of the latter which just about corresponds to the enthalpy difference between diamond and graphite, also in favor of the latter, with one significant difference, namely that the barrier in the carbon system is very high in contrast to the nitric acid trihydrates. However, the phase transition occurs in both systems with differing rates owing to differences in the corresponding barriers. An interesting situation arises in $\alpha$-NAT because the enthalpy of evaporation of $HNO_3$ is lower than in $\beta$-NAT by 32 ± 20 kJ/mol, but this difference is more than compensated by the increased stability of the $H_2O$ network which results in the small, but probably significant enthalpy difference in favor of $\beta$-NAT.

The authors discuss a possible transformation of NAD into NAT. In Tizek et al. (2004) and Grothe et al. (2008), we have performed corresponding laboratory experiments. We may underline that we have observed $\alpha$-NAD transforming into $\beta$-NAD and the respective melting of $\beta$-NAD. However, a phase change from $\alpha$-NAD or $\beta$-NAD into $\beta$-NAT was of extremely low significance and should be kinetically and thermodynamically unfavorable. Also from a stoichiometric point of view, one may conclude that this would include the formation of nitric acid monohydrate or pure nitric acid, which are rather unlikely on thermodynamic grounds. Again a transformation from $\alpha$-NAT into $\beta$-NAT is the most reasonable process, which also fits the observations of Weiss et al. (2016) and Gao et al. (2015).

In this context, one might also remember the AIDA experiments performed by Stetzer et al. (2006) where no NAT but $\alpha$-NAD formation had been observed. The reason might be well involved $HNO_3$ concentrations that were larger than required for NAT formation and so led to NAD. Only recently Iannarelli and Rossi (2015) could show in laboratory experiments that increased $HNO_3$ concentrations exposed to pure ice spontaneously led to NAD.

References

H. Tizek, E. Knözinger, H. Grothe: "Formation and Phase Distribution of Nitric Acid Hydrates in the Mole Fraction Range $x_{HNO_3}$ < 0.25: a combined XRD and IR study"; Physical Chemistry Chemical Physics, 6 (2004), p. 972 - 979.

H. Grothe, H. Tizek, D Waller, D Stokes: "The Crystallization Kinetics and Morphology of Nitric Acid Trihydrate"; Physical Chemistry Chemical Physics, 8 (2006), p. 2232 - 2239.

H. Grothe, H. Tizek, I. Ortega:"Metastable Nitric Acid Hydrates - Possible Constituents of Polar Stratospheric Clouds?"; Faraday Discussions, 137 (2008), p. 223 - 234.

F. Weiss, F. Kubel, O. Galvez, M. Hölzel, S. F. Parker, P. Baloh, R. Iannarelli, J Rossi, H. Grothe: "Metastable Nitric Acid Trihydrate in Ice Clouds"; Angewandte Chemie - International Edition, 55 (2016), 10; p. 3276 - 3280.

R. Iannarelli, M.J. Rossi: "Heterogeneous Kinetics of $H_2O$, $HNO_3$ and HCl on $HNO_3$ hydrates ($\alpha$-NAT, $\beta$-NAT, NAD) in the range 175-200 K", submitted to Atmos. Chem. Phys. Discuss. (2016), ms. no. : acp-2016-247.

R. Iannarelli, M.J. Rossi: "The mid-IR Absorption Cross Sections of α- and β-NAT ($HNO_3 \cdot 3H_2O$) in the range 170 to 185K and of metastable NAD ($HNO_3 \cdot 2H_2O$) in the range 172 to 182K", J. Geophys. Res. Atmos. (2015), 120, doi:10.1002/2015JD023903.

O. Stetzer, O. Möhler, R. Wagner, S. Benz, H. Saathoff, H. Bunz, O. Indris: "Homogeneous nucleation rates of nitric acid dihydrate (NAD) at simulated stratospheric conditions – Part I: Experimental results", Atmos. Chem. Phys. (2006), 6, 3023–3033.

R.-S. Gao, T. Gierczak, T. D. Thornberry, A. W. Rollins, J. B. Burkholder, H. Telg, C. Voigt, T. Peter and D. W. Fahey: "Persistent Water–Nitric Acid Condensate with Saturation Water Vapor Pressure Greater than That of Hexagonal Ice", J. Phys. Chem. A (2016), 120, 9, p. 1431–1440

---

## Author Comment (AC1) · 24 Jun 2016

**Author response to comments of referee #1: "Spectroscopic evidence for large aspherical β-NAT particles involved in denitrification in the December 2011 Arctic stratosphere"**

*Atmos. Chem. Phys. Discuss., doi:10.5194/acp-2016-146, in review, 2016*
W. Woiwode et al.

We would like to thank referee #1 for his/her time and helpful comments and suggestions to improve the manuscript. In the following, we provide the original referee comments (italic letters) followed by our responses. Text added or modified in the revised manuscript is colored in blue.

*Woiwode et al. analyze infrared limb emission spectra that were recorded with the airborne MIPAS-STR instrument during a flight above northern Scandinavia in December 2011. The spectra reveal a "shoulder-like" signature at 820 cm-1 which (in a slightly different manner, i.e., more "peak-like") has already previously been observed in spaceborne infrared PSC observations and has been assigned to beta-NAT particles. Ambient conditions like temperatures around flight altitude during the PSC encounter support the presence of NAT (temperature is above the existence temperatures of ice and STS). Additionally, a local maximum of gaseous HNO3 was detected just below the PSC encounter which could result from nitrification. From simulations with Mie theory, the authors confirm that only with the refractive indices of beta-NAT it is possible to at least roughly reproduce the observed signature at 820 cm-1. But only when considering highly aspherical particle shapes with aspect ratios of 0.1 and 10 in T-matrix computations, a satisfactory agreement between MIPAS-STR measurements and simulations is obtained.*

*General comment*

*This is a very comprehensive study containing a wealth of information, not only the detailed spectral analysis of the PSC signatures, but also concomitant measurements of particle size distributions (in-situ), trace gas analyses, and space-borne observations. In my opinion, the analysis is sound and indeed gives evidence for the presence of highly aspherical beta-NAT particles due to the much better match of the T-matrix simulations compared to those with Mie theory. The manuscript is well-written, and I have listed only a few points that need further clarification. I therefore support the publication in ACP once the following specific comments are addressed.*

We thank referee #1 for this clear summary and encouraging statement.

*Specific comments*

*1) Page 2, line 27: Here and at some other places in the manuscript (in particular at page 10, line 23ff) no mention is made that there are two IR spectroscopically different modifications of NAD, the low-temperature alpha-NAD and the high-temperature beta-NAD phase (Grothe et al., 2004). The employed optical constants from Niedziela et al. (1998) closely correspond to the alpha-NAD spectrum shown in Fig. 6 in Grothe et al. (2004). In beta-NAD, the v2 (NO3-) band is slightly shifted to higher wavenumbers (811 cm-1, Table 1 in Grothe et al., 2004) and, judging from Fig. 6 (in Grothe et al. 2004), has a higher intensity compared to alpha-NAD. As far as I know, optical constants for beta-NAD have not yet been retrieved, and maybe the small wavenumber shift of the v2 (NO3-) band in beta-NAD would not lead to a better match with the measured MIPAS-STR signature at 820 cm-1 compared to alpha-NAD, but the existence of the high-temperature beta-NAD modification should at least be acknowledged. In annealing experiments, Grothe et al. (2004) observed that the low-temperature alpha-NAD modification first transformed into beta-NAD at about 200 K, and that decomposition of beta-NAD into beta-NAT and NAM then only occurred at a considerably higher temperature. This finding should also be addressed in the discussion on page 10, line 25.*

*Grothe, H., Myhre, C. E. L., and Tizek, H., Vibrational spectra of nitric acid dehydrate (NAD), Vib. Spectrosc., 34, 55-62, 2004.*

We thank referee #1 for this hint and modified the manuscript as follows:

P2/L27: As discussed by Grothe et al. (2004), the spectroscopic data of NAD used in these studies closely corresponds with the α-NAD modification. Furthermore, another metastable high-temperature modification β-NAD has been identified by Grothe et al. (2004) in laboratory experiments at temperatures above ~200 K, which decomposes into β-NAT and NAM (nitric acid monohydrate) at considerably higher temperatures.

P8/L4-5: The signatures of NAD are simulated using the refractive indices by Niedzela et al. (1998), which closely correspond with spectroscopic data of α-NAD (Grothe et al., 2004).

P10/L23-27: While α-NAD shows a similar spectral signature with weaker amplitude centred at 808 cm$^{-1}$ (Niedziela et al., 1998, Grothe et al., 2004), the signature is not capable of reproducing the residual dip slightly below 820 cm$^{-1}$ (see Sect. 3.5). The same is expected for the high-temperature modification β-NAD, which was characterized by Grothe et al. (2004) in laboratory experiments and shows a similar spectral signature centred at 811 cm$^{-1}$. Furthermore, the observations indicate temperatures close to the threshold temperature of β-NAT and slightly too warm for NAD under stratospheric conditions.

P19/L31: Grothe, H., Myhre, C. E. L., and Tizek, H., Vibrational spectra of nitric acid dihydrate (NAD), Vib. Spectrosc., 34, 55-62, 2004.

*2) Page 3, lines 4/5: Here and/or in the later discussion on page 13, lines 17-19, one could also explicitly mention that Grothe et al. (2006) observed a variety of morphologies for the beta-NAT particles, depending on the growth conditions, supporting the argumentation that a simplified shape assumption might contribute to the discrepancies between the simulations and the observation.*

We modified as follows:

P3/L5: under laboratory conditions and obtained highly aspherical particles with different morphologies depending on the growth conditions.

P13/L19: This is supported by the experiments of Grothe et al. (2006), resulting in highly aspherical β-NAT particles (i.e. platelets and needles).

*3) Page 8, line 4: See above, the data from Niedziela et al. refer to the signatures of alpha-NAD.*

See above.

*4) Page 8, line 14/15: So size distribution A is a bi-modal log-normal fit to the FSSP-100 observation? One could state this more clearly instead of writing "resembles approximately the size distribution ".*

We clarified as follows:

P8/L14-15: The bimodal size distribution A (Fig. 9b, red) was adjusted manually to match the size distribution derived from the FSSP-100 observations in terms of shape and condensed HNO$_3$.

*5) Page 12, line 21-24, discussion of Fig. 13: I am wondering whether one could elaborate the effect of particle asphericity on the emission/absorption and scattering of the particles more clearly. From the compilation of refractive indices plotted in Fig. 13 c-f it is clear that the "shoulder-like" signal at*

*820 cm-1 can only be reproduced by beta-NAT. But also large spherical beta-NAT particles would produce some sort of shoulder-like signal around that wavenumber from the interplay of "peak-like" emission and "steplike" scattering contribution. The important message is that only highly aspherical particles reproduce the correct amplitude of the measured signal. Obviously, the shape dependency is predominantly related to the scattering contribution, because the AR=1.0 and AR=0.1 scenarios without scattering only showed smaller differences. I would propose to include a more fundamental plot that shows the wavenumber-dependent ensemble-averaged absorption/emission and scattering cross sections for different aspect ratios, so that the reader immediately gets an impression about the change of these basic quantities with particle shape.*

We thank referee #1 for this hint and included the absorption and scattering cross-sections of the optimized scenario for AR=1.0 and 0.1.

P12/L28: Figure 13g and 13h show the ensemble-averaged absorption and scattering cross-sections of β-NAT for the considered size distribution for AR=0.1 and AR=1.0, which determine the absorption/emission and scattering characteristics of the simulated particles. The T-Matrix scenario with AR=0.1 shows a much stronger peak in the absorption cross-section and a stronger step in the scattering cross-section in the spectral window around 820 cm$^{-1}$ when compared to the Mie scenario, which together result in the characteristic "shoulder-like" signature in the simulated spectrum. Furthermore, the AR=0.1 scenario shows considerably higher values of the scattering cross-section towards higher wavenumbers, resulting in a relatively flat baseline of the simulated spectrum towards the upper end of channel 1. In the AR=0.1 scenario, higher absorption and scattering cross-sections in channel 2 result in higher radiances in the corresponding simulated spectrum.

Figure 13:

[Figure]

P39/L5: (g) and (h): ensemble-averaged absorption and scattering cross-sections of β-NAT for the discussed AR=0.1 and AR=1.0 scenarios.

To point out that the absorption cross-sections determine also the emission characteristics of the particles, we modified as follows:

P1/L31: absorption/emission and scattering characteristics

P12/L17: the absorption and emission characteristics of the particles

P12/L28: absorption/emission and scattering

P12/L31: absorption and emission

P13/L7: due to the net emission

P16/L13: absorption/emission and scattering characteristics

*6) Page 12, lines 25-27: See above, no distinction between alpha- and beta-NAD.*

From the modifications in P2/L27, P8/L4-5 and P10/L23-27 it should be clear now that the data used in this work and similar previous studies closely correspond with α-NAD.

*7) Page 16, line 3-4: Is there any assessment of the influence of particle asphericity on the Mie theory-inferred diameters of the FSSP measurements? There is some discussion of this issue in the conclusion section, but I would propose to directly mention it here where the size distributions from the MIPAS-STR simulations and in-situ observations are compared.*

We added:

P16/L5: We mention that Borrmann et al. (2000) investigated the effects of spheroids with AR=0.5 on FSSP observations. Similar to the infrared observations discussed here, the results were close to corresponding Mie calculations. However, the effects of highly aspherical particles on the interpretation of FSSP measurements are uncertain and might explain this discrepancy.

*8) Conclusions section: I would like to see a statement/analysis whether there is a certain size threshold above which one can safely infer shape information for the beta-NAT particles from the signature at 820 cm-1. In the introduction, the authors refer e.g. to previous MIPAS-Envisat PSC observations of beta-NAT particles with smaller radii. Here, a "peak-like" rather than a "shoulder-like" signature at 820 cm-1 was observed, probably due to the reduced amount of scattering. When neglecting the scattering source function, however, the shape influence was observed to be less pronounced (Sect. 3.5). So I am wondering whether there are size limitations for the identification of highly aspherical particles from the MIPAS observations.*

We agree and analysed the transition from a "peak-like" to a "shoulder-like" signature starting with the simplified size distribution B1:

P14/L32: We furthermore perform a sensitivity study based on the scenario involving the simplified size distribution B1 to investigate the effect of decreasing mode radii on the observed spectral signatures when the total volume of β-NAT is kept constant. The results are reported in Appendix B and show that the transition from a "shoulder-like" to a "peak-like" signature occurs for AR=0.1 and the considered mode width at a mode radius of ~3.0 μm. For a mode radius of 1.0 μm, a "peak-like" signature is found in agreement with a corresponding Mie simulation. The results show furthermore, that a modified "shoulder-like signature along with further changes in the simulated spectra results for spherical particles with a mode radius of 3.0 μm.

P16/L20: Sensitivity calculations involving a simplified size distribution show that for AR=0.1 the transition from a "shoulder-like" to a "peak-like" signature occurs at a mode radius of ~3.0 μm. A developed "peak-like" signature as discussed by Höpfner et al. (2006a) is found for a mode radius of 1.0 μm, which is almost identical to the corresponding Mie simulation. Furthermore, a corresponding Mie simulation with a mode radius of 3.0 μm shows that a modified "shoulder-like" signature along with further changes in the modelled spectra can be simulated for spherical particles using the discussed size distribution.

P18/L4: Appendix B

The goal of the sensitivity study discussed in the following is to identify an approximate size threshold for particles with AR=0.1 for the transition from a "peak-like" (compare Höpfner et al., 2006a) to a "shoulder-like" signature in the spectral region around 820 cm$^{-1}$. Corresponding Mie calculations for spherical particles (AR=1.0) are shown for comparison. Starting point for the simulations is the simplified size distribution B1 (1-modal, r=4.8 µm, see Fig. 9a and Table 1, scenario 57g). Sensitivity calculations involve the same total volume of β-NAT (i.e. condensed HNO$_3$) and mode radii of 3.0 µm and 1.0 µm, respectively (Fig. 19).

The results show that for AR=0.1 the spectral signature around 820 cm$^{-1}$ becomes increasingly "peak-like" for mode radii decreasing from 4.8 µm to 1.0 µm (Fig. 20a, 20c, and 20e, blue). While for r=4.8 µm the signature shows a characteristic "shoulder-like" pattern, a superposition of a "shoulder-like" and a "peak-like" signature results for r=3.0 µm. A developed "peak-like" signature as discussed by Höpfner et al. (2006a) is found for r=1.0 µm, and the simulated spectra are almost identical to the corresponding Mie scenario for both channels (Fig. 20e and 20f) except for slightly higher radiances below ~860 cm-1 in for the AR=0.1 scenario. Finally, the Mie calculations show that a modified "shoulder-like" signature around 820 cm$^{-1}$ along with further differences from the AR=0.1 scenario can be modelled for spherical particles with r=3.0 µm.

P45/new Figure 19:

[Figure]

Figure. 19. Size distributions used for sensitivity simulations discussed in Appendix B. The total volume of condensed β-NAT is constant (corresponding with 8.4 ppbv of gas-phase equivalent HNO$_3$) and the mode with is 1.35 in all cases. For the binned size distribution used in the AR=0.1 scenario with r=1.0, the particle number density had to be scaled to 0.17342 cm$^{-3}$ to match 8.4 ppbv of gas-phase equivalent HNO$_3$.

P46/new Figure 20:

[Figure]

Figure. 20. Sensitivity simulations investigating the effect of decreasing mode radii on the simulated spectral signatures of β-NAT particles for a T-Matrix scenario with AR=0.1 (blue) and a corresponding Mie scenario (AR=1.0, red). (a) and (b): large particle mode, size distribution B1. (c) and (d): intermediate particle mode, size distribution B1a. (e) and (f): small particle mode, size distribution B1b. Numbers above (a), (c) and (e) indicate corresponding mode radii and particle number densities (see Fig. 19). 'TM' corresponds with T-matrix simulation and 'SP' with spheroid (numbers indicate AR).

*Technical corrections*

*1) Page 2, line 11: typo in NO3-*

Done

*2) Page 5, line 11: CALIOP should be replaced by CALIPSO .*

Done

*3) Page 9, line 27: I suppose it is meant "by the simulated scattering of radiation", i.e., delete "to".*

Done. We furthermore added at P9/L27: into the field-of-view.

*4) Page 10, line 19: aspherical particles*

Done

*5) Page 11, lines 18, 23, 24: Please check the units for the given radiances. Shouldn't it be W cm-2 sr-1 cm?*

Done. We furthermore corrected units at:

P4/L4: 11-19·$10^{-9}$ W cm$^{-2}$ sr$^{-1}$ cm

P9/L31: ~25·$10^{-9}$ W cm$^{-2}$ sr$^{-1}$ cm

*6) Page 14, line 23: "the first mode is of minor importance"*

Done

*7) Page 16, line30: "of condensed HNO3."*

Done

---

## Author Comment (AC2) · 24 Jun 2016

**Author response to comments of referee #2: "Spectroscopic evidence for large aspherical β-NAT particles involved in denitrification in the December 2011 Arctic stratosphere"**

*Atmos. Chem. Phys. Discuss., doi:10.5194/acp-2016-146, in review, 2016*
W. Woiwode et al.

We would like to thank referee #2 for his/her time and helpful comments and suggestions to improve the manuscript. In the following, we provide the original referee comments (italic letters) followed by our responses. Text added or modified in the revised manuscript is colored in green.

*The authors provide a nice analysis of Fourier transform IR spectroscopic measurements in an Arctic PSC in December 2011. The measurements are analyzed under a variety of radiative transfer situations and with a variety of possible PSC size distributions. The authors convincingly demonstrate the signatures of β-NAT in the data and provide good justification for including the presence of a lower cloud layer and a size distribution with a median radius near 5 µm in the second mode. The paper is well written and should be published. I have three somewhat major comments.*

We thank referee #2 for this concise summary and encouraging statement.

*1) For those not up on the latest PSC literature. How does β-NAT differ from other NAT? What are the other NAT forms? I assume there is an α-NAT. Is there a γ-NAT?*

There are two NAT modifications which are relevant under atmospheric conditions, the thermodynamically stable β-NAT modification and the metastable α-NAT modification. We added the following information:

P2/L4-5: β-NAT is the only nitric acid hydrate known to be thermodynamically stable in condensed state under the conditions of the polar winter stratosphere at temperatures below ~195 K (Hanson and Mauersberger, 1988). The metastable modification α-NAT is another potential PSC constituent at temperatures below ~190 K and transforms irreversibly into β-NAT at higher temperatures (Tizek et al., 2004, and references therein).

*2) What is the temperature history of the β-NAT particles sampled? Since the measurement is in December do the presence of these particles provide any information on the possible formation of NAT through heterogeneous nucleation? Did the temperature fall below Tice in the previous days so the nucleation pathways would come through ice? I realize the authors may not wish to pursue this in an already long paper, but it is a natural question which people will be curious about and it may not take much work.*

We agree that this aspect, which was investigated by Molleker et al. (2014), is interesting and should be discussed. We added the following:

P4/L33: Molleker et al. (2014) calculated backward trajectories for large particles sampled during the discussed flight and found temperatures close to the frost point ~20 h before the flight. While the model temperatures were too warm for ice nucleation, ice particles might have nucleated during lee-wave-induced cooling above Greenland not resolved by the model. Therefore, it is unclear whether the observed particles have nucleated heterogeneously from ice and/or according to a different mechanism.

*3) The discussion of the differences between the condensed hno3 mixing ratio from the various possible size distributions from the different instruments is confusing. Specifically: In section 4, is there an explanation for why the in situ particle size measurements overestimate the condensed phase nitric*

*acid by about a factor of 2 compared to the MIPAS measurements and compared to what may be realistically expected at these altitudes? Even the MIPAS measurements of ~9 ppbv are not consistent with the HNO3 gas phase deficit shown in Fig. 18, which appears to be about 2-3 ppbv.*

We agree that the comparisons of condensed and gas-phase $HNO_3$ are somewhat difficult to follow. The reason for the higher amounts of condensed $HNO_3$ derived from the in situ measurements might be an overestimation of the total volume of condensed $HNO_3$ due to the assumption of spherical particles. Enhancements of gaseous and condensed $HNO_3$ versus unperturbed conditions are derived from the $O_3$-$HNO_3$ correlations in Figure 8. No $HNO_3$ deficit is found around and below flight altitude. Therefore, the observed excess gas-phase and condensed $HNO_3$ must have originated from higher altitudes. We clarified these aspects as discussed below.

*15.8-9 "The bimodal size distribution A corresponds with 18.2 ppbv gas-phase equivalent HNO3 . . . from the FSSP-100 observations, which corresponds to 18.5 ppbv of gas-phase equivalent HNO3". By gas phase equivalent, I assume the authors mean the gas phase mixing ratio should all the condensed phase hno3 be converted to gas phase? If this is the case please state it more explicitly.*

P15/L4: Gas-phase equivalent $HNO_3$ is calculated for pressure and temperature at flight altitude and corresponds with the volume mixing ratio of gaseous $HNO_3$ added to the gas-phase if the particles would evaporate instantaneously.

*This discussion of condensed hno3 is the most confusing of the paper. What is excess hno3, and condensed excess hno3? Excess to what? I assume excess to some predetermined gas phase mixing ratio determined without particles involved. It is not clear how the gray and black hno3 profiles differ in Fig. 18. From the profiles I guess that the black profile is without particles, but this is not stated.*

*The authors conclude that the hno3 in the MIPAS determined size distributions, ~8-9 ppbv, is consistent with the gas phase hno3, but would this not mean that all the hno3 is condensed? If I understand Fig. 18 correctly there should be about 2 ppbv hno3 in the particle phase, implying that this is most consistent with CLAMS. The authors should rewrite this section to clarify this discussion.*

We thank the referee for pointing out this weakness of the discussion and realize that Figures 8 and 18 require more explanation. Figures 18a to 18c represent the individual gas-phase mixing ratio profiles corresponding with the time interval of the PSC encounter (13:39-14:19 UTC, compare Figs. 2, 6 and 18) and used for the correlation shown in Figure 8. Excess $HNO_3$ corresponds with $HNO_3$ exceeding the $O_3$-$HNO_3$ correlation under unperturbed conditions (i.e. outside the polar vortex) and results from condensation, sedimentation and evaporation of $HNO_3$-containing particles (other sources and sinks of $HNO_3$ and $O_3$ are neglected).

From the profiles of the individual gases shown in Figure 18 alone, excess $HNO_3$ (or a $HNO_3$ deficit) cannot be estimated. Excess (deficit) $HNO_{3(g)}$ is derived by comparing the $O_3$-$HNO_3$ correlation during the PSC encounter with the extra-vortex correlation (Figure 8). Excess $HNO_3$ can be both, gas-phase $HNO_{3(g)}$ and $HNO_{3(s)}$ condensed in the PSC exceeding the extra-vortex correlation (Fig. 8). Since all vortex data points in Figure 18 match or exceed the extra-vortex correlation, any further $HNO_{3(s)}$ condensed in the PSC particles has to be excess $HNO_3$, too.

The comparison of excess $HNO_{3(g)}$ in the nitrification layer peak with the particle size distributions aims at putting the amounts of condensed $HNO_{3(s)}$ corresponding with the size distributions into a perspective (e.g. assuming that the nitrification layer resulted from instantaneous evaporation of a PSC layer comparable to the PSC layer present around flight altitude).

For clarification, we modified the manuscript as follows:

[revised manuscript text omitted]

*17.5-10. Here is a plausible explanation for the discrepancy between the in situ and MIPAS estimates of condensed hno3, but this possibility should be described in section 4 where this discrepancy is discussed.*

We agree that it makes sense to discuss this aspect in the section 4 and shifted P17/L5-10 shifted to P16/L5. Together with modifications related to the comments by referee #2 (blue), P16/L4-5 now read as follows:

P16/L4-5: in-situ observations. We mention that Borrmann et al. (2000) investigated the effects of spheroids with AR=0.5 on FSSP observations. Similar to the infrared observations discussed here, the results were close to corresponding Mie calculations. However, the effects of highly aspherical particles on the interpretation of FSSP measurements are uncertain and might explain this discrepancy.

The larger particle sizes derived from the FSSP-100 and CDP measurements using the Mie theory are not necessarily in contradiction with the radiative transfer simulations of the MIPAS-STR observations discussed here when interpreted as maximum dimensions of highly aspherical particles. For example, elongated spheroids with extreme aspect ratios can easily span lengths of several tens of microns while having relatively small individual particle volumes. Evidence of particles with sizes of this magnitude is provided by CIP shadow cast images recorded during the Arctic winter 2009/10 (Molleker et al. 2014).

The CLaMS simulation suggests …

P17/L5-10 replaced by: The discrepancies between the particle size distributions derived from the MIPAS-STR observations and the in situ observations might be due to the fact that spherical particles were assumed in the evaluation of the in situ observations. On the other hand, the particle sizes derived from the in situ observations may be reconciled with the simulations of the MIPAS-STR observations when interpreted as the maximum dimensions of highly aspherical particles.

*Minor comments:*

*Figure 2. Why are there no MIPAS circles above the flight altitude of the Geophysica?*

The MIPAS-STR observations are performed in downward-looking mode (limb-mode). Additional upward-viewing measurements are also performed to obtain limited information (mainly column) of atmospheric constituents above the flight path. As a consequence of the measurement geometries, only the downward-looking observations have tangent points along the line-of sight, which are plotted

in Figure 2 (i.e. sampling grid). Upward-viewing measurements could be included into Figure 2 by replacing the vertical axis by vertical viewing angle, which however would not be useful in this context.

The retrieved vertical profiles and vertical cross-sections (Fig. 6, 7, 17 and 18) correspond with the employed retrieval grid (vertical spacing of 0.5 km in the discussed range, see Woiwode et al., 2012). From the combination of the information included in the limb observations and the upward-viewing geometries, some vertically resolved information can be obtained at and slightly above the flight path. However, the vertical resolution decreases rapidly above flight altitude. Therefore, the Figures showing retrieval results (temperature and trace gases) can include data points above the flight altitude.

*7.16, "The close agreement of the temperatures measured by MIPAS-STR, the UCSE data and the ECMWF data with calculated TNAT around flight altitude suggests that the observed PSC was composed of β-NAT." Does this correspondence really specifically indicate β-NAT, or just NAT? I have never seen such temperatures specifically target a particular type of NAT. Also what is β-NAT and how does it differ from ??? What is/are the other option(s)? This should be discussed in the introduction.*

See above: The existence temperature of β-NAT is higher than for the other atmospheric relevant modification α-NAT and the other potential candidates STS and ice. The potential relevance of α-/β-NAD is addressed in the response to referee #1.

*8.11 "In Figure 9a, the particle size distributions derived by Molleker et al. (2014) from the FSSP-100 and CDP observations (black and magenta, respectively) during the PSC encounter are shown." This figure shows observations I think, not a derived size distribution. Please clarify.*

We used "derived" to point out that the in situ probes do not directly measure particle size distributions, but particle size distributions are derived from the measured forward scattering signal involving the Mie model or other models. Thereby, the choice of the model, the particle geometry and the particle type has consequences for the resulting size distribution.

*9.27 "into the simulated spectra by to the simulated" Fix.*

Done.

*10.10-32 and Fig. 10. Do we really need to put the reader through all these Mie calculations since the temperatures are clearly too warm for STS and ice? It seems a waste of space to include Figs 10 e)-h) and the discussion of ice and STS, for all the reasons listed at the bottom of page 10. Just state these reasons up front as to why STS and ice are not considered.*

We would like to include these plots to show how different refractive indices affect the radiative transfer simulations using the Mie model. In context of the comment by H. Grothe and M. J. Rossi, we furthermore included Mie calculations for α-NAT.

*Comment related to most of the Figs 10 -16. A little more effort on the legends in the figure captions would be appreciated. I recognize the space limitations, but the readers do not know what NATcoa means. Why the coa? Nor do the readers know what SP means. I can guess SP10 may mean an aspect ratio of 10 and SP01 an aspect ratio of 0.1, but no directions are provided. Why SP? Use the space to indicate the type of calculation, the type of particle, the aspect ratio, and if possible the median diameter of the second mode*

We agree that the legends should be explained in more detail. We mention that information and references for "NATcoa" are given at P8/L2-4 and that the numbers in the legends provide the link to Table 1, which provides all details of the simulations. We modified as follows:

P8/L4: Hereafter, we refer with β-NAT to the "NATcoa" refractive indices by Biermann et al (1998).

P36/L6: 'NATcoa' corresponds with β-NAT and 'Mie' with Mie simulation (AR=1.0).

P37/L5, P38/L5, P41/L6 and P42/L6: 'NATcoa' corresponds with β-NAT, 'TM' with T-matrix simulation and 'SP' with spheroid (numbers indicate AR).

P39/L3: 'NATcoa' corresponds with β-NAT, 'Mie' with Mie simulation, 'TM' with T-matrix simulation and 'SP' with spheroid (numbers indicate AR).

P40/L4: 'NATcoa' corresponds with β-NAT, 'Mie' with Mie simulation, 'TM' with T-matrix simulation and 'SP' with spheroid (numbers indicate AR).

*13.13-19. Does the fact that the particles may not be homogeneous, throughout the layer sampled, play a role in diluting/spreading the signal? Are there some minimum number of NAT particles required for the signature?*

Horizontally inhomogeneous particle size distributions along the line of sight would modify the observed spectra. However, the spectra measured in different horizontal directions during the PSC encounter (compare Figure 1 with Figures 4a and 4b) show only moderate variations and suggest a relatively homogeneous PSC in horizontal direction.

The minimum particle number densities required for a significant signature depend on (i) the mode radius/radii and width/s of the size distribution, (ii) the vertical thickness of the PSC and (iii) the scattering contribution in the signal as a consequence of the tropospheric cloud scenario and surface temperature and emissivity (see section 3.6). Therefore, the minimum number densities required for a significant signature around 820 cm$^{-1}$ due to the $\nu_2$ mode of β-NAT are strongly case-dependent.

---

## Author Comment (AC3) · 24 Jun 2016

**Author response to interactive comment of H. Grothe and M. J. Rossi: "Spectroscopic evidence for large aspherical β-NAT particles involved in denitrification in the December 2011 Arctic stratosphere"**

*Atmos. Chem. Phys. Discuss., doi:10.5194/acp-2016-146, in review, 2016*
W. Woiwode et al.

We would like to thank H. Grothe and M. J. Rossi for their interesting comments, remarks and suggestions to improve the manuscript. In the following, we provide the original short comment (italic letters) followed by our responses. Text added or modified in the revised manuscript is colored in red.

*The investigation by Woiwode et al. (2016) represents an improved and updated reinterpretation of spectroscopic data on the existence of NAT particles in the Artic lower stratosphere. However, the whole evidence is based on the symmetric nitrate deformation band $v_2=\delta_s(NO_3^-)$, which is in general the weakest band of the overall mid-infrared spectrum of nitric acid hydrates. Also, the differences between the $v_2$ bands of NAT (α, β) and NAD (α, β) are rather small. These differences may also be caused by different crystal structures or different morphologies and textures of the very same phase. This might be worth to be mentioned in the text. Unless concomitant changes in other parts of the mid IR spectrum are not conclusively observed, minor changes in one of the IR absorptions alone constitute weak evidence for the reasons cited above.*

We initially excluded α-NAT due to too warm temperatures, a weaker signature at 820 cm$^{-1}$, and the fact that this constituent has not yet been confirmed in the stratosphere from FTIR observations so far to our best knowledge. Furthermore, in our opinion the positions of the $v_2$ bands of $NO_3^-$ in α/β-NAD are significantly different from α/β-NAT in context of high-resolution infrared observations. We agree that a more detailed discussion of the spectral signatures would be helpful. Furthermore, we would like to clarify that the identification of β-NAT is not limited only to the $v_2$ band of $NO_3^-$, but also to characteristic spectral patterns in the entire channels 1 (725-990 cm$^{-1}$) and 2 (1150-1350 cm$^{-1}$) of MIPAS-STR.

- The signature around 820 cm$^{-1}$ attributed to the $v_2$ mode of $NO_3^-$ in β-NAT well exceeds the signal-to noise of the measurements. Therefore, the fact alone that this band has a smaller amplitude than other bands of the same substance does not mean that it is not suitable for identification.

- The spectral signature at 820 cm$^{-1}$ is particularly well suited for identification, since it is located in a region of the spectrum weakly populated by gaseous absorbers and where thermal emission is high relative to other regions covered by the MIPAS-STR observations due to the increase of atmospheric thermal emission towards lower wavenumbers. Furthermore, it represents a sharp feature, having a significant amplitude in both the imaginary and real parts of the refractive indices and resulting in characteristic signatures in the absorption and scattering cross-sections of large highly aspherical β-NAT particles (see response to referee #1). The amplitudes of the 820 cm$^{-1}$ signature of β-NAT in the refractive index imaginary and real parts used here are significantly higher than the corresponding signatures in available refractive indices of α-NAT and (α-)NAD.

- The simulations of the MIPAS-STR spectra using the refractive indices of β-NAT reproduce the signature attributed to the $v_2$-band of β-NAT in both (i) spectral position and (ii) amplitude and therefore provide a strong hint at β-NAT. As discussed by Höpfner et al. (2006a), many studies consistently report a spectral position at ~809 cm$^{-1}$ for the $v_2$-band of $NO_3^-$ in (α-)NAD, which is significantly different from the position at ~820 cm$^{-1}$ consistently reported in the

literature for both α-NAT and β-NAT. We are not familiar with any study suggesting a signature at 820 cm$^{-1}$ almost identical to β-NAT in both in position and amplitude for a nitric acid hydrate other than β-NAT.

- The identification of β-NAT not only is based on the $v_2$ band of $NO_3^-$ in the spectral region around 820 cm$^{-1}$, but also on refractive index patterns in the entire spectral ranges of MIPAS-STR channels 1 and 2. These cover the broad $v_3$ asymmetric stretch band of $NO_3^-$ around ~1330 cm$^{-1}$ and the $v_2$ symmetric umbrella mode of $H_3O^+$ around ~1200 cm$^{-1}$, which result in broad, weak and unspecific signatures in the simulated spectra. While in the Mie simulations, the channel 2 spectra are almost identical for the considered nitric acid hydrate modifications, notable differences are found in the comparison with STS and ice. Within the simulations of nitric acid hydrates, the combination of the specific signature around 820 cm$^{-1}$ of β-NAT with the unspecific signatures in channel 2 and further non-specified distinct patterns (see notable differences between Mie scenarios of β-NAT, α-NAT (see below) and NAD in channel 1 above 830 cm$^{-1}$) provides a characteristic fingerprint, which becomes more pronounced for aspheric particles.

We included also α-NAT in the analysis and modified the manuscript as follows (text modified in context of the comments by referee #1 is colored in blue.):

P1/L20-22: We analyse polar stratospheric cloud (PSC) signatures in airborne MIPAS-STR (Michelson Interferometer for Passive Atmospheric Sounding – STRatospheric aircraft) observations in the spectral regions from 725 to 990 cm$^{-1}$ and 1150 to 1350 cm$^{-1}$ under conditions suitable for the existence of nitric acid trihydrate (NAT) above northern Scandinavia on 11 December 2011.

P1/L24: $v_2$ symmetric deformation mode of $NO_3^-$ in β-NAT

P2/L27: To our best knowledge, α-NAT has not been identified in infrared field observations so far.

P7/L17-18: α-NAT, STS and ice are unlikely candidates due to too high temperatures.

P8/L4: For simulating the signatures of α-NAT, we use the refractive indices for α-NAT aerosols by Richwine et al. (1995), which show a more developed signature around 820 cm$^{-1}$ than the corresponding faint signature in the refractive indices for α-NAT by Toon et al. (1994).

P9/L13: Mie calculations of spherical β-NAT, α-NAT, NAD, STS and ice particles

P9/L14: β-NAT, α-NAT, NAD,

P10/L8-11 (discussion of residuals further refined): For α-NAT (Fig. 10c and 10d), NAD (Fig. 10e and 10f), and STS (Fig. 10g and 10h), similar residuals are found for both size distributions when compared to β-NAT. For these species, more "step-like" residual signatures are found in the spectral region around 820 cm$^{-1}$ for both size distributions when compared to β-NAT.

P10/L1511 (discussion of residuals further refined): a "step-like" dip

P10/L19: we exclude α-NAT, NAD,

P10/L21-27:

[revised manuscript text omitted]

P39 (refractive indices of α-NAT included in Figures 13c to 13f):

[Figure]

P39/L5: of β-NAT, α-NAT, NAD, STS and ice.

*The authors have detected highly aspherical NAT particles, which are in accordance with lab experiments (XRD, ESEM) and Avrami calculations by Grothe et al. (2006) and Tizek et al. (2004). The main aspect ratio is 1:10 or 10:1. The authors should however indicate clearly if they assign these ratios to needles or platelets? Are optical calculations able to distinguish between these alternatives?*

We would like to clarify that out of the scenarios analyzed for β-NAT (AR=0.1, 0.5, 1.0, 2 and 10), the best overall agreement is found for elongated spheroids with AR=0.1. Thereby, the choice of the extreme aspect ratios was motivated by the work of Grothe et al. (2006) (see P3/L4). However, more extreme AR values for both elongated and oblate spheroids may further improve the agreement. The modeled spectral signatures show different sensitivities to changes in AR for elongated and oblate particles due to the different contributions from absorption and scattering. However, from the scenarios analyzed we cannot tell which particle geometry would ultimately lead to the best agreement, if the AR values would be modified sufficiently. Further aspects modifying the signal are the exact particle shape (e.g. presence of edges and flat surfaces) and potential shape-dependent orientation effects. So in summary, within the scenarios analyzed best agreement is found for elongated spheroids with AR=0.1. However, we cannot exclude that other combinations of particle geometries, aspect ratios and potential shape-dependent orientation effects would favor a different particle geometry.

We modified the manuscript as follows:

P1/L28-29: Within the scenarios analyzed, the best overall agreement is found for elongated spheroids with AR=0.1.

P11/L16: The choice of the AR values is motivated by the laboratory experiments by Grothe et al. (2006), resulting in needles and platelets with similar proportions depending on the crystallization conditions.

P16/L28: Further combinations of particle shapes (e.g. including edges and flat surfaces), aspect ratios and potential shape-dependent orientation effects might further improve the agreement of simulations and observations.

*Only recently Weiss et al. (2016) have shown that α-NAT has a much lower heterogeneous nucleation barrier than β-NAT, where ice acts as the heterogeneous nucleus. In a lab experiment, HNO3 was deposited from the gas phase onto an ice film and the outcome was a pure α-NAT film on top of the ice substrate – similar findings are also reported by Gao et al. (2015). This is important in view of the here presented findings, since Grothe et al (2006) found long needles with an aspect ratio of 1:10 only in the presence of ice. On the other hand platelets with a much smaller aspect ratio were found in experiments without ice. So the straightforward conclusion would be that the history of the here observed NAT particles should have involved a heterogeneous nucleation step of α-NAT on ice and a subsequent transformation of α-NAT into β-NAT. It would be interesting to note if the presence of ice can be confirmed by limb emission spectra.*

See comment to referee #2: Particle backward trajectories by Molleker et al. (2014) suggest that model temperatures were too warm for ice nucleation. Furthermore, temperatures around flight altitude were clearly too warm for ice. However, ice particles might have nucleated during lee-wave-induced cooling above Greenland not resolved by the model. Therefore, ice might have been involved in the nucleation of the observed particles.

We modified the manuscript as follows:

P16/L19: The fact that best agreement is found for highly elongated particles might hint on a heterogeneous nucleation of the particles involving ice and a subsequent phase transition from α-NAT to β-NAT (Grothe et al., 2006, Iannarelli et al., 2016, Weiss et al. 2016, and references therein). While the temperatures at flight altitude were too warm for ice and α-NAT and model temperatures do not support a previous ice nucleation, Molleker et al. (2014) suggest that ice particles might have nucleated previously during lee-wave-induced cooling above Greenland not resolved by the model, enabling an ice-induced nucleation of NAT.

P20/L16: Iannarelli, R. and Rossi, M. J.: Heterogeneous Kinetics of $H_2O$, $HNO_3$ and HCl on $HNO_3$ hydrates (α-NAT, β-NAT, NAD) in the range 175–200 K, Atmos. Chem. Phys. Discuss., doi:10.5194/acp-2016-247, in review, 2016.

P22/L19: Weiss, F., Kubel, F., Gálvez, O., Hoelzel, M., Parker, S.F., Iannarelli, R., Rossi, M.J. and 1094 Grothe, H.: Metastable Nitric Acid Trihydrate in Ice Clouds, Angewandte Chemie I.E., 55, 1095 3276-3280, 2016; doi: 10.1002/anie.201510841, 2016.

*Additionally, Tizek et al. (2004) found a stabilization of α-NAT in ice matrices. The phase transition was so much hindered by the ice that at about 200 K α-NAT could still be observed for several hours. It might be worth to check for the presence of α-NAT in the here presented spectra. Iannarelli and Rossi (2016) have estimated an enthalpy difference of 6 ± 20 kJ/mol between α- and β-NAT in favor of the latter which just about corresponds to the enthalpy difference between diamond and graphite, also in favor of the latter, with one significant difference, namely that the barrier in the carbon system is very high in contrast to the nitric acid trihydrates. However, the phase transition occurs in both systems with differing rates owing to differences in the corresponding barriers. An interesting situation arises in α-NAT because the enthalpy of evaporation of HNO3 is lower than in β-NAT by 32 ± 20*

*kJ/mol, but this difference is more than compensated by the increased stability of the H2O network which results in the small, but probably significant enthalpy difference in favor of β-NAT.*

While the temperatures at flight altitude were too warm for the persistence of ice and α-NAT under atmospheric conditions, we now include Mie simulations for α-NAT with spherical geometry (which would be supported by the work of Grothe et al., 2006) to test for this species (see above). The results of the simulations do not support the presence of large spherical α-NAT particles.

*The authors discuss a possible transformation of NAD into NAT. In Tizek et al. (2004) and Grothe et al. (2008), we have performed corresponding laboratory experiments. We may underline that we have observed α-NAD transforming into β-NAD and the respective melting of β-NAD. However, a phase change from α-NAD or β-NAD into β-NAT was of extremely low significance and should be kinetically and thermodynamically unfavorable. Also from a stoichiometric point of view, one may conclude that this would include the formation of nitric acid monohydrate or pure nitric acid, which are rather unlikely on thermodynamic grounds. Again a transformation from α-NAT into β-NAT is the most reasonable process, which also fits the observations of Weiss et al. (2016) and Gao et al. (2015).*

We thank the authors of the comment for pointing out this aspect and delete the statement at P10/L26, suggesting a phase transition from NAD into NAT. As discussed above, a potential phase transition from α-NAT to β-NAT is now mentioned in the conclusions.

*In this context, one might also remember the AIDA experiments performed by Stetzer et al. (2006) where no NAT but α-NAD formation had been observed. The reason might be well involved HNO3 concentrations that were larger than required for NAT formation and so led to NAD. Only recently Iannarelli and Rossi (2015) could show in laboratory experiments that increased HNO3 concentrations exposed to pure ice spontaneously led to NAD.*

This very interesting aspect might help to better understand the results by Stetzer et al. (2006) and design further experiments.